# MODEL-FREE OFFLINE REINFORCEMENT LEARNING WITH ENHANCED ROBUSTNESS

**Chi Zhang**[1]**, Zain Ulabedeen Farhat**[1]**, George K. Atia**[1,2]**, Yue Wang**[1,2]

[1] Department of Electrical and Computer Engineering    [2] Department of Computer Science
University of Central Florida
Orlando, FL 32816, USA
{chi.zhang,zainulabedeen.farhat,george.atia,yue.wang}@ucf.edu

## ABSTRACT

Offline reinforcement learning (RL) has gained considerable attention for its ability to learn policies from pre-collected data without real-time interaction, which makes it particularly useful for high-risk applications. However, due to its reliance on offline datasets, existing works inevitably introduce assumptions to ensure effective learning, which, however, often lead to a trade-off between robustness to model mismatch and scalability to large environments. In this paper, we enhance both aspects with a novel double-pessimism principle, which conservatively estimates performance and accounts for both limited data and potential model mismatches, two major reasons for the previous trade-off. We then propose a universal, model-free algorithm to learn a policy that is robust to potential environment mismatches, which enhances robustness in a scalable manner. Furthermore, we provide a sample complexity analysis of our algorithm when the mismatch is modeled by the $l_\alpha$-norm, which also theoretically demonstrates the efficiency of our method. Extensive experiments further demonstrate that our approach significantly improves robustness in a more scalable manner than existing methods.

## 1 INTRODUCTION

Traditional reinforcement learning (RL) (Sutton & Barto, 2018) optimizes an agent's performance through iterative trial-and-error interactions with the environment, and has shown significant success in many areas such as video games (Wei et al., 2022; Liu et al., 2022a). However, such an online learning scheme can be costly or unsafe in real-world applications. For instance, in domains including autonomous driving (Kiran et al., 2021), stock market trading (Kabbani & Duman, 2022), and healthcare (Yu et al., 2021), poor decisions can have significant consequences, making extensive explorations impractical. To address them, *offline RL* has been developed (Lange et al., 2012; Levine et al., 2020), enabling agents to learn from pre-collected datasets, offering a more reliable framework.

Since offline RL relies heavily on pre-collected datasets, the quality of these datasets largely determines performance. It is hence unclear whether satisfactory performance can be achieved for complex problems with a relatively limited dataset. In this context, two key challenges in improving offline RL performance have been studied. The first is **scalability**—the ability to handle large-scale problems. Without real-time interaction, learning an effective policy for large-scale problems from a limited dataset, which may not fully cover the entire state-action space, can be challenging. Recent research has focused on improving scalability by adapting model-free algorithms (Shi et al., 2022; Yan et al., 2022; Laroche et al., 2019; Fujimoto et al., 2019; Ghasemipour et al., 2021; Kumar et al., 2019; Wu et al., 2019; Siegel et al., 2020) and leveraging function approximation techniques (Ross & Bagnell, 2012; Liu et al., 2020; Xie et al., 2021a; Yin et al., 2021a; Xie & Jiang, 2021; Jiang & Huang, 2020). However, due to the complexity of large environments, many of these approaches assume that the dataset sufficiently represents the full deployment environment, typically presuming that the deployment environment is identical to the one from which the data was collected.

However, this assumption can be too restrictive. Static datasets only capture the environment at the time of data collection, but real-world applications frequently face environmental uncertainty due to perturbations or non-stationarity. This mismatch between the data collection and deployment

environments, commonly known as the *sim-to-real gap* (Zhao et al., 2020), can cause significant performance degradation during deployment. Therefore, it is crucial to enhance the **robustness** of offline RL to ensure that the learned policies can perform reliably in the presence of such uncertainties. A promising solution is to adapt robust RL frameworks (Iyengar, 2005; Nilim & El Ghaoui, 2004) to the offline setting, as explored recently in (Shi & Chi, 2022; Blanchet et al., 2023). However, these methods often come at the cost of scalability. Due to their inherent structure, robust RL methods typically rely on dynamic planning, which requires knowledge of the full transition dynamics, and are predominantly model-based. This necessitates learning and storing a complete transition model, which is resource-intensive (Zhang et al., 2021a) and limits scalability for large-scale problems.

Recognizing the limitations of current methods and the challenges posed by large-scale problems and model uncertainty, a trade-off between robustness and scalability becomes apparent. Enhancing one typically comes at the expense of the other. This naturally leads to the following question:

*Can we develop a unified framework that enhances both scalability and robustness in offline RL?*

In this paper, we address this question by presenting a model-free algorithm to learn a policy that is both robust to model uncertainty and scalable to large-scale problems. Our method introduces a principle of double pessimism to simultaneously address two key sources of uncertainty: (1) the uncertainty arising from inaccurate estimations due to the underexplored datasets, and (2) model mismatch between the data collection and deployment environments. We then propose a streamlined conceptual framework, design a model-free algorithm, and provide the first theoretical guarantee of convergence and robustness of our approach. Our contributions can be summarized as follows.

- **A Double-Pessimism Principle for Offline RL with Model Mismatch.** We begin by framing the challenge of enhancing robustness in offline RL within an offline robust RL framework, where an uncertainty set captures potential environmental mismatches. To solve offline robust RL in a scalable manner, we propose the double-pessimism principle that does not require transition kernel estimations. This principle maintains a conservative estimate of robust performance, obtained directly from data collection without requiring model estimation. We then introduce the first model-free pessimistic robust Q-learning algorithm. Our algorithm optimizes performance under model mismatch using an offline dataset, while offering greater memory efficiency and more scalability than previous methods.

- **First and Near-Optimal Model-Free Algorithm for Offline Robust RL.** We provide a rigorous sample complexity analysis for our model-free double-pessimistic robust Q-learning algorithm under the widely used $l_\alpha$-norm uncertainty set. Our analysis shows that, given a dataset satisfying the partial coverage condition (to be introduced later), our algorithm can identify an optimal robust policy with near-optimal sample complexity, comparable to that of model-based offline robust RL and model-free offline non-robust RL. This represents the first sample complexity analysis for model-free robust offline RL, demonstrating its applicability to large-scale problems that require high data efficiency.

- **Numerical Experimental Verification of Enhanced Robustness.** We conduct extensive numerical experiments to demonstrate the improvements in robustness achieved by our algorithms in both simulated environments (Archibald et al., 1995) and real physics-based Classic Control problems (Brockman et al., 2016). In each case, our algorithm consistently outperforms existing methods in handling model uncertainty, showcasing its enhanced ability to maintain stable performance across a wide range of environmental perturbations. Moreover, our approach demonstrates superior scalability stemming directly from our model-free algorithm design, as shown by its effectiveness in solving more complex Classic Control problems with robustness guarantees, which have proven difficult or unsolvable for previous model-based robust methods.

## 2 PRELIMINARIES

### 2.1 FINITE-HORIZON MARKOV DECISION PROCESS (MDP)

A finite-horizon MDP is represented by $\mathcal{M} = \left(\mathcal{S}, \mathcal{A}, H, P \triangleq \{P_h\}_{h=1}^H, r \triangleq \{r_h\}_{h=1}^H\right)$, where $\mathcal{S}$ and $\mathcal{A}$ are the finite state and action spaces of size $S$ and $A$, respectively, and $H$ is the horizon length.

The probability transition kernel $P_h : \mathcal{S} \times \mathcal{A} \to \Delta(\mathcal{S})$ and the reward function $r_h : \mathcal{S} \times \mathcal{A} \to [0,1]$ are defined at each step $h$ ($1 \le h \le H$). At each step $h$, the agent starts in state $s_h$, takes action $a_h$, transitions to the next state $s_{h+1}$ according to the transition kernel $P_{h,s_h,a_h}$, and receives a reward $r_h(s_h, a_h)$. This process terminates after $H$ steps when the agent reaches state $s_{H+1}$.

A policy $\pi = \{\pi_h\}_{h=1}^{H}$ defines the strategy for selecting actions in different states, where $\pi_h : \mathcal{S} \to \Delta(\mathcal{A})$ specifies the probability distribution over actions at step $h$. The performance of an agent following a policy $\pi$ is measured by the value function $V^{\pi,P} = \{V_h^{\pi,P}\}_{h=1}^{H}$, where

$$V_h^{\pi,P}(s) \triangleq \mathbb{E}_{\pi,P}\left[ \sum_{t=h}^{H} r_t(s_t, a_t) \,\Big|\, s_h = s \right]. \tag{1}$$

The expectation is taken over the trajectory $\{s_h, a_h, r_h\}_{h=1}^{H}$ generated by executing the policy $\pi$ and transitioning according to the transition kernel $P$: $a_h \sim \pi_h(s_h)$ and $s_{h+1} \sim P_{h,s_h,a_h}$.

## 2.2 INFINITE-HORIZON MDP

An infinite-horizon MDP is defined as $\mathcal{M} = \big(\mathcal{S}, \mathcal{A}, P, r, \gamma\big)$, where both the transition kernel $P$ and the reward function $r$ are stationary and do not change over time. The discount factor $\gamma < 1$ ensures the finiteness of the accumulated reward over an infinite horizon.

Due to its stationary nature, it suffices to consider only stationary policies $\pi : \mathcal{S} \to \Delta(\mathcal{A})$, which specify the action-selection probabilities over the action space. The value function $V^{\pi,P}$ of a policy $\pi$ with transition kernel $P$ is defined as

$$V^{\pi,P}(s) \triangleq \mathbb{E}_{\pi,P}\left[ \sum_{t=1}^{\infty} \gamma^t r_t(s_t, a_t) \,\Big|\, s_0 = s \right]. \tag{2}$$

## 2.3 ROBUST MDP

A finite-horizon robust MDP (RMDP) is specified by $\big(\mathcal{S}, \mathcal{A}, H, \mathcal{P} = \{\mathcal{P}_h\}, r\big)$, and an infinite-horizon RMDP is denoted by $\big(\mathcal{S}, \mathcal{A}, \mathcal{P}, r, \gamma\big)$, where $\mathcal{P}$ is a set containing some transition kernels, named the uncertainty set. At each step, the environment transitions to the next state following an arbitrary kernel belonging to the uncertainty set, instead of a fixed one as in non-robust MDPs. In this paper, we consider the $(s, a)$-rectangular uncertainty set (Wiesemann et al., 2013), where $\mathcal{P}$ is independently defined for each state-action pair, with $\bigotimes$ denoting the Cartesian product:

$$\mathcal{P}_h = \bigotimes_{(s,a) \in \mathcal{S} \times \mathcal{A}} \mathcal{P}_{h,s,a} \text{ (finite-horizon)}, \quad \mathcal{P} = \bigotimes_{(s,a) \in \mathcal{S} \times \mathcal{A}} \mathcal{P}_{s,a} \text{ (infinite-horizon)}. \tag{3}$$

The performance of a policy in an RMDP is evaluated based on its worst-case value function over all the instances in the uncertainty set. Specifically, the finite-horizon robust value functions $V^\pi = \{V_h^\pi\}_{h=1}^{H}$ and the infinite-horizon robust value functions $V^\pi$ are defined as

$$V_h^\pi(s) \triangleq \inf_{P \in \mathcal{P}} V_h^{\pi,P}(s) \text{ (finite-horizon)}, \quad V^\pi(s) \triangleq \inf_{P \in \mathcal{P}} V^{\pi,P}(s) \text{ (infinite-horizon)}$$

where the infimum is taken over the uncertainty set of transition kernels. For a given initial state distribution $\rho \in \Delta(\mathcal{S})$, we write the expected robust performance as

$$V_1^\pi(\rho) \triangleq \mathbb{E}_{s_1 \sim \rho}[V_1^\pi(s_1)] \text{ (finite-horizon)}, \quad V^\pi(\rho) \triangleq \mathbb{E}_{s \sim \rho}[V^\pi(s)] \text{ (infinite-horizon)}. \tag{4}$$

The goal of an RMDP is to learn a policy that optimizes the worst-case performance, or equivalently, the robust value functions. Such a policy is referred to as an optimal robust policy:

$$\pi^* = \{\pi_h^*\} \triangleq \arg\max_\pi V_1^\pi(\rho), \text{ (finite-horizon)}, \tag{5}$$

$$\pi^* \triangleq \arg\max_\pi V^\pi(\rho), \text{ (infinite-horizon)}. \tag{6}$$

## 3 FORMULATION: ENHANCING ROBUSTNESS AND SCALABILITY

In this section, we develop our formulation, where we utilize RMDPs to formulate the offline RL problem against model mismatch.

In the offline setting, the dataset is collected under a fixed environment $P$ (referred to as the nominal kernel) by executing some behavior policy $\mu$. However, due to factors such as non-stationarity, unexpected perturbations, or adversarial attacks, the deployment environment may differ from $P$. To account for this model deviation and improve robustness, we construct an uncertainty set by perturbing the nominal kernel and aim to learn the optimal robust policy. Specifically, following (Xu & Mannor, 2010; Xu et al., 2010; Derman et al., 2021; Kumar et al., 2023), we define the uncertainty set (of $(s, a)$-pair) for modeling environmental perturbations as:

$$\mathcal{P}_{h,s,a} = \{P_{h,s,a} + q \in \Delta(\mathcal{S}) : q \in \mathcal{Q}_{h,s,a}\} \quad \text{(finite-horizon)}, \tag{7}$$
$$\mathcal{P}_{s,a} = \{P_{s,a} + q \in \Delta(\mathcal{S}) : q \in \mathcal{Q}_{s,a}\} \quad \text{(infinite-horizon)}, \tag{8}$$

for some set $\mathcal{Q}_{h,s,a}, \mathcal{Q}_{s,a}$ containing the possible model perturbations, and aim to learn the optimal robust policy for the corresponding RMDPs. This will not only provide an optimized lower bound on performance when the deployment environment lies within the uncertainty set, but also improves the robustness to model uncertainty (Pinto et al., 2017).

### 3.1 FINITE-HORIZON

In the finite-horizon setting, the dataset $\mathcal{D}$ consists of $K$ episodes each of length $H$. These episodes are independently generated based on a certain behavior policy $\mu$ and the nominal kernel $P$:

$$\mathcal{D} = \{(s_1^k, a_1^k, r_1^k, ..., s_H^k, a_H^k, r_H^k, s_{H+1}^k)_{k=1,...,K}\}, \tag{9}$$

where $a_i^k \sim \mu(\cdot|s_i^k)$, $s_{i+1}^k \sim P_{i,s_i^k,a_i^k}$, and the initial state $s_1^k \sim \rho$.

Since the dataset is collected by a fixed policy under a single nominal environment, there exists a distribution shift between the data distribution, and the distribution induced by the optimal policy and the worst-case kernel. To guarantee that a provable efficient algorithm can be designed based on the dataset, we adopt a popular assumption on the distributional mismatch between the dataset distribution and the occupancy measure induced by the optimal policy $\pi^*$, as in (Shi & Chi, 2022).

**Assumption 1** (Robust single-policy concentrability). *The behavior policy $\mu$ satisfies that*

$$C^* \triangleq \max_{(s,a,P',h)\in\mathcal{S}\times\mathcal{A}\times\mathcal{P}\times[H]} \frac{d_{P',h}^{\pi^*}(s,a)}{d_{P,h}^{\mu}(s,a)} < +\infty, \tag{10}$$

*where $d_{P,h}^{\pi}$ is the occupancy distribution induced by policy $\pi$ and transition kernel $P$ at step $h$.*

In Assumption 1, we only require that the dataset covers the state-action pairs that are visited by the optimal policy, known as the partial coverage condition (Rashidinejad et al., 2021).

Our goal is then to learn an $\epsilon$-optimal policy $\hat{\pi}$ for the RMDP with the uncertainty set defined as in equation 3 and equation 7, such that

$$V_1^{\pi^*}(\rho) - V_1^{\hat{\pi}}(\rho) \leq \epsilon. \tag{11}$$

### 3.2 INFINITE-HORIZON

In the infinite-horizon setting, the offline dataset contains a single trajectory of length $T$ obtained by executing a behavior policy $\mu$ under the nominal kernel $P$:

$$\mathcal{D} = \{s_1, a_1, r_1, s_2, ..., s_T\}, \tag{12}$$

where $s_1 \sim \rho$, $a_i \sim \mu(\cdot|s_i)$ and $s_{i+1} \sim P_{s_i,a_i}$. For the infinite horizon setting, we adopt the following two assumptions on the behavior policy.

We first adopt the partial coverage assumption in (Blanchet et al., 2023; Wang et al., 2024c).

**Assumption 2.** *The behavior policy $\mu$ satisfies*

$$C^* \triangleq \max_{(s,a,P') \in \mathcal{S} \times \mathcal{A} \times \mathcal{P}} \frac{d_{P'}^{\pi^*}(s,a)}{d_P^{\mu}(s,a)} < +\infty, \tag{13}$$

*where $d_P^{\pi}$ denotes the occupancy distribution induced by policy $\pi$ and transition kernel $P$.*

We make an additional assumption on the behavior policy as follows.

**Assumption 3.** *The behavior policy $\mu$ is stationary, and the induced Markov chain under the nominal kernel is uniformly ergodic.*

**Remark 1.** *This assumption is commonly adopted in prior works (Wang et al., 2020; Yan et al., 2022; Li et al., 2020; Wang & Zou, 2020), as it ensures that the dataset includes all state-action pairs covered by the **behavior policy**, provided the dataset size exceeds a certain threshold. This assumption is required since the dataset consists of a single Markovian trajectory. When the dataset contains i.i.d. samples from the occupancy distribution $d_P^{\mu}$, as in (Wang et al., 2024c; Li et al., 2022), such an assumption can be removed.*

Our goal is then to find an $\epsilon$-optimal policy $\hat{\pi}$ through $\mathcal{D}$ for the RMDP with the uncertainty set defined in equation 3 and equation 8, such that

$$V^{\pi^*}(\rho) - V^{\hat{\pi}}(\rho) \leq \epsilon. \tag{14}$$

## 4 DOUBLE-PESSIMISM PRINCIPLE

In this section, we introduce our model-free algorithm for learning an optimal robust policy from an offline dataset. As we mentioned, two major challenges in offline RL are the two sources of uncertainty: one arising from the limited and under-explored dataset, and the other from the mismatch between the data collection and target environments. We aim to develop a unified double-pessimism principle to address both of them.

As suggested by previous studies on offline RL, e.g., (Rashidinejad et al., 2021; Li et al., 2022; Shi et al., 2022; Yan et al., 2022; Wang et al., 2024c), the uncertainty arising from the dataset can be addressed using a single-pessimism principle. This involves introducing a penalty term $b_n$, which depends on the visitation frequency of each state-action pair, to penalize less frequently visited pairs. By doing so, we obtain a conservative estimate of the value function under the nominal kernel.

However, addressing the uncertainty arising from model mismatch is particularly challenging, especially with a model-free approach. Most previous robust RL studies require that the estimation of the worst-case transition, $\sigma_{\mathcal{P}}(V) \triangleq \min_{p \in \mathcal{P}} pV$, be unbiased. This can be satisfied when the agent can freely generate data as needed (e.g., (Wang et al., 2023d; Liu et al., 2022b; Wang et al., 2023c;b)), yet is impractical in offline settings. To address this issue, we argue that another pessimism principle can be adopted, and that learning a policy robust to model mismatch does not require an unbiased estimator or an accurate solution to the worst-case. As long as the estimator provides a (not too pessimistic) lower bound on the worst-case, it is sufficient to account for the uncertainty due to model mismatch and still learn a robust policy. We therefore propose a model-free estimator that lower bounds $\sigma(V)$ to produce a conservative estimation as follows.

**Definition 1.** *For the uncertainty set $\mathcal{P}_{s,a}$, a function $\kappa$ is referred to as a model-mismatch penalty function if for any non-negative vector $V$ and a sample $s' \sim P_{s,a}$ from the nominal kernel,*

$$\mathbb{E}[V(s') - \kappa_{s,a}(V)] \leq \sigma_{\mathcal{P}_{s,a}}(V). \tag{15}$$

A universal design of the penalty function $\kappa$ is provided in Appendix C. Such a penalty function ensures that at each step, the updated estimate represents a lower bound on the true worst-case scenario, resulting in a conservative estimation and enhancing robustness. We note that this additional pessimism may result in a more conservative policy, as the algorithm will estimate the robust value function more pessimistically. However, we argue that as long as the pessimism level is not too large, the learned policy will not be too conservative, maintaining a satisfactory performance and enhancing the robustness. More importantly, calculation of $\kappa$ does not require any information of the model, but can be done in a data-driven and model-free fashion.

We then combine the two pessimism principles together, to develop our double-pessimism algorithm based on the Q-learning algorithm. For each sample $(s, a, s')$, we update the $Q$ table by

$$Q(s, a) \leftarrow \qquad (1 - \eta)Q(s, a) + \eta \left( r(s, a) + \gamma V(s') - \underbrace{\gamma \kappa_{s,a}(V)}_{\text{model mismatch}} - \underbrace{b_n(V)}_{\text{limited dataset}} \right). \qquad (16)$$

As we will show later, such an update rule incorporating the double-pessimism principle ensures that our estimation is conservative, and can effectively tackle the uncertainty in offline robust RL. More importantly, such an update rule does not require any information on the transition model, and hence can be adapted in a model-free manner and is more suitable for large-scale problems.

Based on this, we develop our model-free offline algorithms for both finite and infinite horizon cases. In the following sections, we present these algorithms and develop their sample complexity analysis.

## 5 DOUBLE-PESSIMISM Q-LEARNING FOR FINITE-HORIZON MDPS

Adopting the double-pessimism principle, we propose our algorithm for finite-horizon MDPs.

---

**Algorithm 1** Double-Pessimism Q-Learning for finite-horizon RMDPs.

---

**Input:** $\mathcal{D}$, target success probability $1 - \delta$, uncertainty set radius $R$, penalty function $\kappa$
**Initialize:** $Q_h(s, a) = 0$, $N_h(s, a) = 0$, $V_h(s) = 0$, $\forall s, a, h$
**for** $k = 1, \ldots, K$ **do**
    Sample a trajectory $\{s_h, a_h, r_h\}_{h=1}^H$ from $\mathcal{D}_\mu$
    **for** $h = 1, \ldots, H$ **do**
        $N_h(s_h, a_h) \leftarrow N_h(s_h, a_h) + 1$; $n \leftarrow N_h(s_h, a_h)$; $\eta_n \leftarrow \frac{H+1}{H+n}$
        $b_n \leftarrow c_b \sqrt{\frac{H^3 \log^2(SAKH/\delta)}{n}}$
        $Q_h(s_h, a_h) \leftarrow (1 - \eta_n)Q_h(s_h, a_h) + \eta_n \left\{ r_h(s_h, a_h) + V_{h+1}(s_{h+1}) - \kappa_{h,s_h,a_h}(V_{h+1}) - b_n \right\}$
        $V_h(s_h) \leftarrow \max \left\{ V_h(s_h), \max_a Q_h(s_h, a) \right\}$
    **end for**
    $\widehat{\pi}_h^k(s) \leftarrow \arg\max_a Q_h(s, a)$, $\forall s, h$
**end for**
$\widehat{\pi}_h(s) \leftarrow \widehat{\pi}_h^K(s)$, $\forall s, h$
**Output:** $\widehat{\pi} = \{\widehat{\pi}_h\}$

---

In our algorithm, the term $\kappa$ is for conservative estimation of the worst-case performance within the uncertainty set, while the term $b$ addresses the pessimism of the limited dataset. We track the visitation count of each state-action pair and construct the penalty term $b$ based on these counts. As the dataset visits a pair more frequently, the associated uncertainty decreases and $b$ decreases.

**Remark 2.** *Our algorithm design is universal and works for any uncertainty set models, as long as we have a penalty function $\kappa$ satisfying equation 15, which is provided in Appendix C. However, since $\kappa$ for different models requires individual studies, we mainly derive our theoretical analysis for the $l_\alpha$-norm models (Kumar et al., 2023; Derman et al., 2021):*

$$\mathcal{P}_{h,s,a} = \left\{ q \in \Delta(\mathcal{S}) : \|q - P_{h,s,a}\|_\alpha \leq R_{h,s,a} \right\}. \qquad (17)$$

*We again emphasize that our double-pessimism principle and algorithm design can be extended further to other uncertainty set models. We provide a detailed discussion on $\kappa$ in Appendix C.*

Next, we develop our theoretical results for $l_\alpha$-norm sets. We first show that equation 15 is satisfied by our design, and the algorithm results in a conservative estimation of the robust value function.

**Lemma 1.** *For the $l_\alpha$-norm uncertainty set, set the penalty function $\kappa$ as*

$$\kappa_{h,s,a}(V) \triangleq R_{h,s,a} \min_{w \in \mathbb{R}} \|we - V\|_\beta, \qquad (18)$$

*where $\beta = \frac{1}{1-\frac{1}{\alpha}}$ is the Hölder conjugate of $\alpha$, and $e = (1, 1, ..., 1) \in \mathbb{R}^S$. Then, equation 15 is satisfied. Moreover, it holds that for all $(k, h, s) \in [K] \times [H] \times \mathcal{S}$,*

$$V_h(s) \leq V_h^{\widehat{\pi}_h^k}(s) \leq V_h^{\star}(s). \tag{19}$$

The lemma provides a concrete construction of the penalty function for the $l_\alpha$-norm model. More importantly, our model-free estimator and algorithm result in pessimistic estimations of robust value functions, tackling both uncertainty sources. In our next result, we show that our double-pessimism principle is effective in learning the optimal robust policy from the mismatched offline dataset.

**Theorem 2.** *For the $l_\alpha$-norm uncertainty set, and any $\delta \in (0, 1)$, suppose that the behavior policy $\mu$ satisfies Assumption 1. When $T \triangleq HK > \tilde{\mathcal{O}}(SC^\star)$, the policy $\widehat{\pi}$ returned by Algorithm 1 satisfies*

$$V_1^{\pi^\star}(\rho) - V_1^{\widehat{\pi}}(\rho) \leq \tilde{\mathcal{O}}\left(\sqrt{\frac{H^6 SC^\star}{T}}\right) \tag{20}$$

*with probability at least $1 - \delta$. Here, $\pi^*$ is the optimal robust policy w.r.t. a (possibly) relaxed $l_\alpha$-norm uncertainty set (see Appendix C.2 for detailed discussion). $f(T) = \tilde{\mathcal{O}}(g(T))$ means that $|f(T)| \leq C \cdot g(T) \cdot (\log g(T))^k$ for some constants $C > 0$ and $k \geq 0$, when $T$ is sufficiently large.*

Our algorithm is the first model-free algorithm for offline RL under model mismatch with sub-optimality gap analysis. The sub-optimality gap we obtain in the previous result further implies that we can learn an $\epsilon$-optimal policy as long as the size of the offline dataset $T$ exceeds

$$\underbrace{\tilde{\mathcal{O}}\left(\frac{H^6 SC^\star}{\epsilon^2}\right)}_{\epsilon\text{-dependent}} + \underbrace{\tilde{\mathcal{O}}(SC^\star)}_{\text{burn-in cost}}. \tag{21}$$

Note that in the sample complexity, the second term, referred to as the burn-in cost, is a universal constant that does not depend on $\epsilon$, while the first term asymptotically depends on $\epsilon$. When $\epsilon$ becomes smaller, the first term dominates the overall complexity, resulting in an asymptotic complexity of $\tilde{\mathcal{O}}\left(\frac{H^6 SC^\star}{\epsilon^2}\right)$. A more detailed discussion of the complexity will be provided in Section 7.

**Remark 3.** *When the radius $R$ is small, it holds that $\mathbb{E}[V(s') - \kappa_{s,a}(V)] = \sigma_{\mathcal{P}_{s,a}}(V)$ (see Theorem 1 in (Kumar et al., 2023)), hence Algorithm 1 converges to the optimal robust policy w.r.t. the original uncertainty set. For general uncertainty set and corresponding penalty function $\kappa$, Algorithm 1 may converge to the optimal robust policy w.r.t. a relaxed uncertainty set, as the estimation may be inaccurate. However, robustness can still be enhanced due to the additional pessimism. See Appendix C for further discussion.*

## 6 DOUBLE-PESSIMISM Q-LEARNING FOR INFINITE-HORIZON MDPs

In this section, we present our algorithm design and analysis for offline RL with infinite-horizon MDPs. Due to space limitation and similarities in algorithm design, the algorithm is deferred to Algorithm 3 in Appendix E.1. The algorithm follows a similar design as the finite-horizon one, where the two terms $\kappa$ and $b$ represent conservative penalties for the double-pessimism principle. Again, our algorithm design is universal, but we develop the sample complexity results only for $l_\alpha$-norm models.

**Theorem 3.** *Consider the $l_\alpha$-norm uncertainty set and any $\delta \in (0, 1)$. Suppose that the behavior policy $\mu$ satisfies Assumption 2 and Assumption 3. Then, the policy $\widehat{\pi}$ returned by Algorithm 3 satisfies*

$$V^\star(\rho) - V^{\widehat{\pi}}(\rho) \leq \tilde{\mathcal{O}}\left(\sqrt{\frac{C^\star S}{T(1-\gamma)^5}} + \frac{C^\star S}{T(1-\gamma)^2} + \frac{C^\star}{T(1-\gamma)^3}\right) \tag{22}$$

*with probability at least $1 - \delta$.*

An $\epsilon$-optimal robust policy can be learned as long as the size of the offline dataset exceeds

$$\tilde{\mathcal{O}}\left(\frac{SC^*}{(1-\gamma)^5 \epsilon^2}\right). \tag{23}$$

This sample complexity matches the results of model-free offline non-robust RL (Yan et al., 2022) without variance reduction techniques, which implies the near-optimality of our method. Compared to model-based offline robust RL (Shi & Chi, 2022; Blanchet et al., 2023), our result matches theirs in terms of $C^\star, S, \epsilon$, but exhibits a higher order dependence on $(1 - \gamma)$. We argue that, in general, model-free algorithms tend to have lower memory requirements but incur higher sample complexity compared to model-based approaches. A more detailed discussion will be provided in Section 7.

# 7 RELATED WORK

## 7.1 COMPARISON WITH PRIOR ARTS

In this section, we compare our work to the most closely related studies for tabular offline robust RL (Shi & Chi, 2022; Blanchet et al., 2023). The results are summarized in Table 1, where we only include the infinite horizon ones. Compared to previous studies, our method offers improved memory and computational complexity, while maintaining comparable sample complexity.

| Reference | Memory complexity | Sample complexity | Computational complexity |
|:---:|:---:|:---:|:---:|
| Our Work | $\mathcal{O}(SA)$ | $\tilde{\mathcal{O}}\left(\frac{SC^\star}{\epsilon^2(1-\gamma)^5}\right)$ | Polynomial |
| (Blanchet et al., 2023) | $\mathcal{O}(S^2A)$ | $\tilde{\mathcal{O}}\left(\frac{S^2C^\star}{\epsilon^2(1-\gamma)^4}\right)$ | NP Hard |
| (Shi & Chi, 2022) | $\mathcal{O}(S^2A)$ | $\tilde{\mathcal{O}}\left(\frac{SC^\star}{\epsilon^2 P_{\min}(1-\gamma)^4}\right)$ | Polynomial |

Table 1: Comparison with offline robust RL works. (Shi & Chi, 2022) is for the KL-divergence set.

First, both related works are model-based, which involves estimating and storing the transition model $\{\hat{P}_{s,a} : (s, a) \in \mathcal{S} \times \mathcal{A}\} \in \mathbb{R}^{S^2A}$. This approach thus requires an additional memory of size $\mathcal{O}(S^2A)$ to store the model, along with $\mathcal{O}(SA)$ space for the number of visited state-action pairs from the dataset. As a result, it becomes inefficient for large-scale problems or environments with complicated transition dynamics. In contrast, our model-free algorithm only requires $\mathcal{O}(SA)$-sized space for the number of visits. Such a reduced memory complexity enables our model-free algorithms to handle large-scale problems, scaling effectively to large-scale or even continuous problems.

In terms of computational complexity, the most related work (Blanchet et al., 2023) requires to solve a non-rectangular RMDP, which is generally NP-hard (Wiesemann et al., 2013). In contrast, our algorithm can be effectively implemented in polynomial time, which is much more practical. Compared to (Shi & Chi, 2022), our algorithm still enjoys lower computational complexity, since the update rule of the model-based approach requires computing the inner product $\hat{P}_{s,a}V$, whereas our model-free approach eliminates this computation and only requires a single vector entry $V(s')$. See Appendix C for a more detailed discussion.

In terms of sample complexity, both of our sample complexity results match the ones for offline non-robust Q-learning without variance reduction, illustrating our data efficiency and near-optimality. Our result improves the dependence on $S$ compared to (Blanchet et al., 2023) under the $l_\infty$-norm uncertainty set, showing the enhanced scalability to large-scale problems. On the other hand, it is the general observation that model-based methods tend to demonstrate better sample complexity in terms of $(1 - \gamma)$ than model-free methods, especially when additional techniques like variance reduction are not employed. Such findings have been widely noted in various settings, for instance, when comparing robust RL with generative models ((Wang et al., 2024a) vs. (Shi et al., 2023)) and non-robust offline RL ((Yan et al., 2022) vs. (Li et al., 2022)).

On a side note, we note that our result for the finite-horizon setting exhibits a higher-order dependence on $H$ (where we set $H = \frac{1}{1-\gamma}$ as the effective horizon in infinite setting). This is due to the non-stationary environment inherent in the finite-horizon setting, which is also consistent with findings from previous studies, such as in (Shi & Chi, 2022).

To summarize, our approach addresses existing gaps in offline RL by enhancing robustness to model mismatch, reducing memory requirements, and providing adaptability to large-scale problems, establishing a state-of-the-art method in the field.

## 7.2 OTHER RELATED WORKS

**Offline RL without model mismatch.** A significant body of offline RL works assumes identical collection and deployment environments. Based on that, many early works further rely on the global coverage assumption, where the behavior policy covers all state-action pairs (Scherrer, 2014; Chen & Jiang, 2019; Munos, 2005; Yin et al., 2021b; Yin & Wang, 2021a; Jiang, 2019; Wang et al., 2019; Liao et al., 2020; Liu et al., 2019; Zhang et al., 2020; Uehara et al., 2020; Duan et al., 2020; Xie & Jiang, 2020; Levine et al., 2020; Antos et al., 2007; Farahmand et al., 2010). This assumption is often too restrictive and unrealistic, as it requires complete coverage of state-action pairs in historical data (Gulcehre et al., 2020; Agarwal et al., 2020a; Fu et al., 2020). A more practical partial coverage setting is later proposed, allowing to learn from a less explored dataset. Under partial coverage, the optimal policy can still be learned by incorporating the pessimism principle to handle dataset uncertainty (Jin et al., 2021; Uehara & Sun, 2021; Xie et al., 2021a;b; Rashidinejad et al., 2021; Zanette et al., 2021; Yin & Wang, 2021b; Shi et al., 2022; Li et al., 2022; Zhan et al., 2022; Wang et al., 2023e; Kumar et al., 2020). Differently, we consider potential model mismatches.

**Robust RL.** Robust RL (Iyengar, 2005; Nilim & El Ghaoui, 2004; Xu & Mannor, 2010) aims to tackle the challenge of model mismatch in RL, by optimizing the worst-case performance over an uncertainty set. Existing work focuses mainly on the online setting (Wang & Zou, 2021; 2022; Wang et al., 2023a; Badrinath & Kalathil, 2021; Dong et al., 2022; Lu et al., 2024; Liu & Xu, 2024a) or with a generative model (Yang et al., 2021; Xu et al., 2023; Panaganti & Kalathil, 2022; Shi et al., 2023; Wang et al., 2024a;b; 2022). Offline robust RL, except for the two mentioned above, either relies on strong assumptions, such as global coverage or absorbing states (Panaganti et al., 2022; Yang et al., 2021), or employs fitted type algorithm designs (Yang et al., 2022; Panaganti et al., 2022; Liu et al., 2023). More importantly, most of them are model-based, while we develop the first model-free algorithm for offline robust RL. Another line of research aims to improve robustness and scalability through function approximation (Liu & Xu, 2024b; Wang et al., 2024a; Ma et al., 2022), yet we focus on the tabular setting to develop a more fundamental understanding of offline RL. Another line of robust RL aims to optimize the performance under the environment from a corrupted dataset collected under the same environment (Yang et al., 2023; Zhang et al., 2021b; 2022), which is different from our setting.

## 8 EXPERIMENTS

We use numerical experiments to demonstrate the advantages of our framework in terms of robustness. We consider two sets of environments: simulated MDPs with controllable transition dynamics and Classic Control environments. More experiments are further provided in Appendix B.

### 8.1 SIMULATION MDPS

We first evaluate the performance of our algorithm on the Garnet problem (Archibald et al., 1995), a randomly generated MDP $\mathcal{G}(a, b, c)$ with $a$ states, $b$ actions, and $c$ branches (see Appendix A for a more detailed description). Both the nominal kernels and reward functions are generated randomly. The uncertainty set is constructed using the $l_\infty$-norm, with the radius $R_{s,a} \in [0.1, 0.5]$.

We first generate a dataset of size $N$ from the nominal kernel and apply our double-pessimism algorithm, with the single-pessimism baseline (Yan et al., 2022), to learn policies. We then compute the robust value functions of the learned policies and plot the difference between these values and the optimal robust value functions, referred to as the optimality gap, in Figure 1. The results are averaged over 10 times, with the maximum and minimum gaps as an envelope around the average value. The results show that our double-pessimism algorithm converges to the true optimal robust value as the dataset size increases, maintaining a lower optimality gap, while the single-pessimism approach results in a larger gap. These findings demonstrate that our double-pessimism principle significantly enhances robustness while remaining model-free and scalable.

### 8.2 CLASSIC CONTROL PROBLEMS

To further demonstrate the improvements in both scalability and robustness offered by our approach, we consider more complex Classic Control tasks from OpenAI Gym (Brockman et al., 2016),

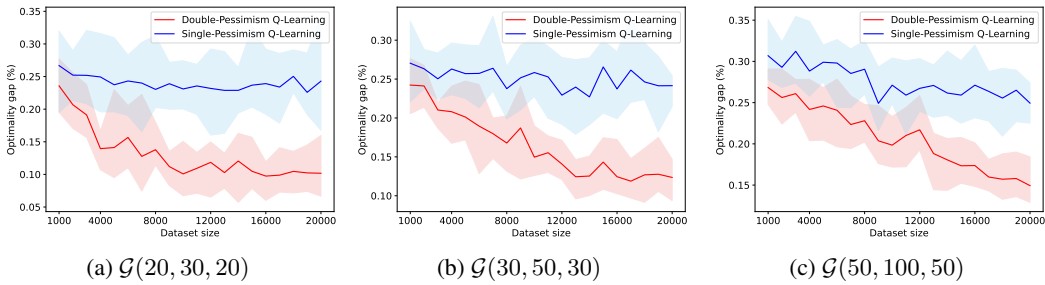

(a) $\mathcal{G}(20, 30, 20)$       (b) $\mathcal{G}(30, 50, 30)$       (c) $\mathcal{G}(50, 100, 50)$

Figure 1: Optimality gaps under different Garnet problems.

specifically MountainCar and CartPole (results are shown in Figure 4 in Appendix). The dynamics of these environments are indirectly controlled by their parameters, e.g., the length of the pole in CartPole, the gravity and the force in MountainCar, and it is of interest to improve the robustness against their uncertainty. Since these model mismatches are hard to model, model-based approaches become ineffective,yet our model-free method remains applicable and effective in such scenarios.

For each dataset generated under the nominal environment with the default parameters, we implemented our algorithm alongside the baseline (Yan et al., 2022) to learn policies. To evaluate the robustness of the learned policies, we test their performance in modified environments with parameter perturbations (Pinto et al., 2017; Wang & Zou, 2021), where we randomly perturbed these parameters within the range of $[-\tau, \tau]$ for 800 trials. As shown in Fig. 2, our double-pessimism algorithm maintains a higher average performance under environment perturbations, demonstrating superior robustness, which aligns with our theoretical findings. This illustrates the enhanced robustness achieved by our approach. Moreover, given the large-scale and complex dynamics of these environments which are difficult for model-based approaches, our model-free algorithm effectively addresses these challenges, further demonstrating the scalability of our method.

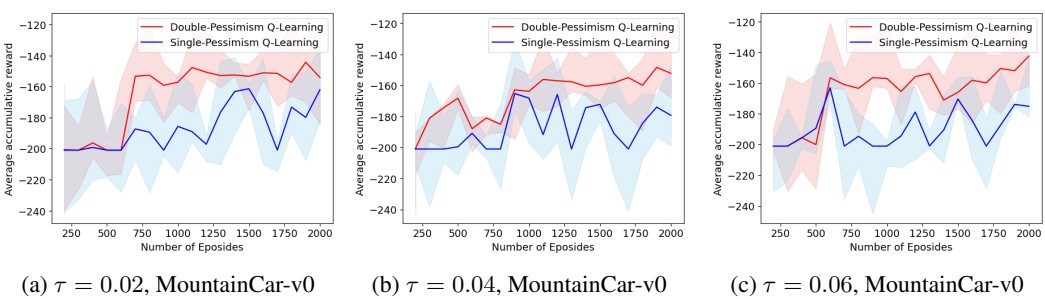

(a) $\tau = 0.02$, MountainCar-v0     (b) $\tau = 0.04$, MountainCar-v0     (c) $\tau = 0.06$, MountainCar-v0

Figure 2: Reward profiles with different parameter perturbations.

## 9 CONCLUSION

We explored offline RL with a focus on improving scalability and robustness simultaneously. We framed the problem as offline robust RL and developed a model-free algorithm to optimize the worst-case performance within an uncertainty set accounting for the possible model mismatch. To address two key challenges—uncertainty from the under explored dataset and model mismatch between data collection and deployment environments—we introduced a double-pessimism principle that conservatively estimates the agent's performance in a model-free manner. Building on this, we designed a universal model-free algorithm that eliminates the need for model estimation, adapts to various uncertainty sets, and scales to large problems. We further analyzed its performance for the widely studied $l_{\alpha}$-norm uncertainty set, showing near-optimal data efficiency of our approach. Our approach significantly improves the robustness, scalability, and efficiency of offline RL compared to existing methods, pushing the boundaries of offline RL research.

ACKNOWLEDGMENT

This work was supported by DARPA under Agreement No. HR0011-24-9-0427 and NSF under Award CCF-2106339. The authors thank the anonymous reviewers whose constructive comments led to substantial improvement to the paper. The authors gratefully acknowledge Xinran Tang at the University of Central Florida for helpful discussions.

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

## A    EXPERIMENTAL SETUP OF SECTION 8

### A.1    GARNET PROBLEMS

For simulated MDP environments, we implement Algorithm 3 on Garnet problems $\mathcal{G}(20, 30, 20)$, $\mathcal{G}(30, 50, 30)$ and $\mathcal{G}(50, 100, 50)$. Here, the branch number denotes the number of states that can be achieved after taking an action. The uncertainty radius $R_{s,a}$ is randomly drawn from a uniform distribution ranging from 0.1 to 0.5 for all state-action pairs. The true robust expected values for the Garnet problems, over a certain state distribution, can be obtained via the model-based robust value iteration method. For each problem, we first generate a stochastic behavior policy with partial coverage over state-action pairs. To obtain a near-optimal stochastic behavior policy, we compute the Q-values for the nominal kernel, and adopt a softmax transformation to assign probabilities for all state-action pairs. The randomness (i.e., optimality) of the behavior policy is controlled via temperature parameter $t_{\mathsf{b}} = 1$. State-action pairs with probabilities $P_{s,a} \leq 0.03$ (for $\mathcal{G}(20, 30, 20)$), $P_{s,a} \leq 0.02$ (for $\mathcal{G}(30, 50, 30)$) and $P_{s,a} \leq 0.01$ (for $\mathcal{G}(50, 100, 50)$) are then excluded to achieve partial coverage. Finally, non-zero elements are re-normalized to maintain a valid probability distribution. By deploying the behavior policy on the nominal kernel, 10 datasets are generated at each dataset size from $T = 1000$ to $T = 20000$. We compared the double-pessimism method with the single-pessimism method in (Yan et al., 2022). We set $\gamma = 0.95$, $C_{\mathsf{b}} = 1 \times 10^{-4}$ and $\delta = 0.02$.

### A.2    CLASSIC CONTROL PROBLEMS

Note in the Classic Control problems, the underlying uncertain environments may not be modeled using our perturbation-based uncertainty set in equation 8, but we can still implement our algorithms to enhance the robustness. We generate the dataset according to a random behavior policy, and implement Algorithm 3 with the radius $R = \tau$. In our experiments, we set $\gamma = 0.95$, $C_{\mathsf{b}} = 1 \times 10^{-4}$ and $\delta = 0.02$. After a policy is learned, we test its performance under a perturbed environment with the parameter randomly generated from $[-\tau, \tau]$ for 800 times, and plot the average performance among them.

## B    ADDITIONAL EXPERIMENT RESULTS

### B.1    COMPARISONS IN TABULAR ENVIRONMENTS

In this section, we include additional experiment results under three simulated environments. Specifically, we consider the Frozen-Lake and Taxi environments from OpenAI Gym (Brockman et al., 2016), and the American Option problem (Panaganti et al., 2022; Shi & Chi, 2022; Zhou et al., 2021). The transition dynamics of these environments can be directly controlled, and we construct $l_\infty$-norm uncertainty sets centered at their nominal kernels. Similarly, we trained our double-pessimism Q-learning together with the single-pessimism baseline, and plotted the optimality gap between the learned and optimal robust value functions. As the results in Figure 3 show, our double-pessimism Q-learning effectively obtains the optimal robust policy, whereas the single-pessimism Q-learning only achieves sub-optimal performance. The results hence indicate that our additional pessimism effectively enhances robustness against model uncertainty, verifying our theoretical results.

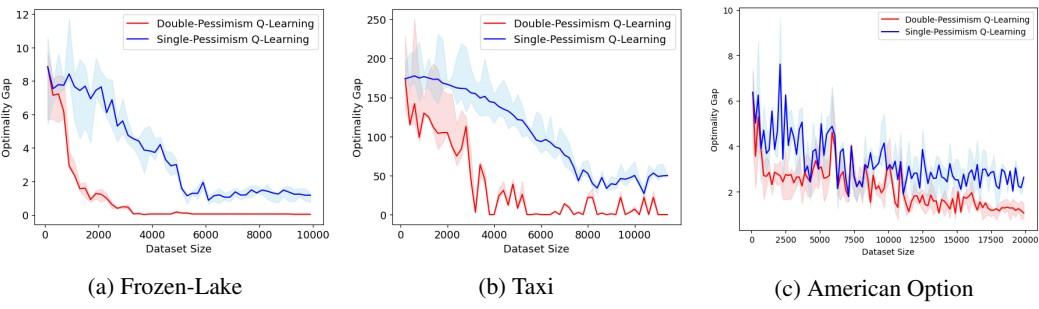

(a) Frozen-Lake                     (b) Taxi                     (c) American Option

Figure 3: Optimality gaps under different Gym environments.

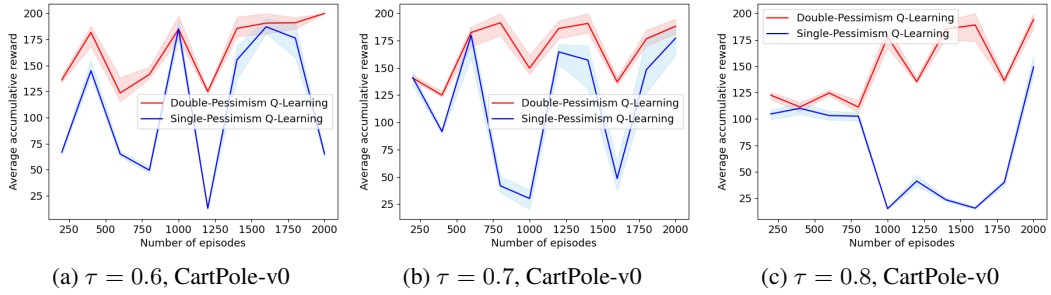

Figure 4: Reward profiles with different parameter perturbations.

## B.2    SCALABLE ALGORITHM WITH FUNCTION APPROXIMATION: DOUBLE-PESSIMISM CQL

In this section, we extend the evaluation of our double-pessimism framework to large-scale problems using function approximation techniques. The algorithms presented earlier (Algorithm 1, Algorithm 3), while model-free, are designed for tabular settings and require memory space of $\mathcal{O}(SA)$ for the $Q$-table, making them less efficient for large-scale applications. To improve scalability, replacing the $Q$-table with low-dimensional function approximations (e.g., neural networks) to reduce memory costs is a widely adopted approach. On the other hand, existing offline RL algorithms like Conservative Q-learning (CQL, (Kumar et al., 2020)) and Implicit Q-learning (IQL, (Kostrikov et al., 2021)), along with others (Ross & Bagnell, 2012; Laroche et al., 2019; Fujimoto et al., 2019; Kumar et al., 2019; Agarwal et al., 2020b; Liu et al., 2020; Jin et al., 2021; Xie et al., 2021a; Yin et al., 2021a; Rashidinejad et al., 2021; Xie & Jiang, 2021; Jiang & Huang, 2020), have focused solely on offline RL **without model mismatch**, resulting in degraded performance when model mismatch is present.

Aiming to enhance both robustness and scalability, we design and evaluate a double-pessimism CQL algorithm, demonstrating that our framework is not limited to tabular settings but can also be integrated with function approximation or deep RL techniques, significantly improving robustness against model mismatch. Specifically, we employ the CQL method to impose pessimism on the limited dataset, and further incorporate an additional penalty term into the robust Bellman operator estimation to effectively mitigate model mismatch. Based on this construction, we can similarly design a double-pessimism CQL algorithm, from which enhanced robustness is expected.

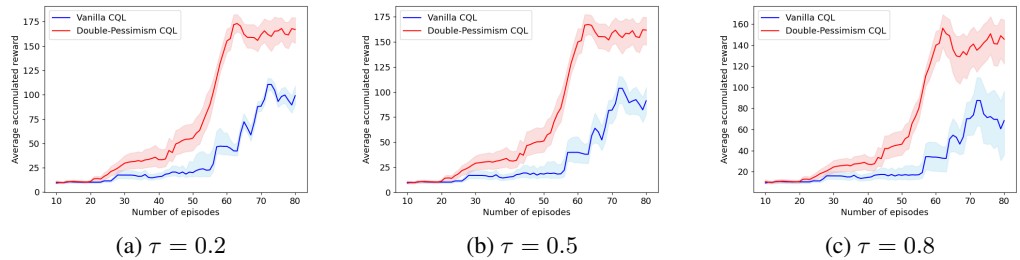

Figure 5: Double-Pessimism CQL vs. Vanilla CQL under CartPole.

To validate the effectiveness of our double-pessimism principle, we compare our double-pessimism CQL with the vanilla single-pessimism CQL under CartPole from OpenAI Gym. The policy is trained in the nominal environment and evaluated in randomly perturbed environments (perturbation radius $\tau$) over 800 trials. The results, shown in Figure 5, display the average performance as solid curves, with envelopes representing standard deviations.

As the results indicate, our double-pessimism CQL consistently outperforms the vanilla CQL in perturbed environments, demonstrating enhanced robustness. This experiment confirms the universal applicability of our double-pessimism framework in improving robustness, regardless of the specific algorithm used. It also highlights the scalability of our approach, which can be integrated with advanced deep offline RL algorithms for large-scale problems using function approximation.

### B.3 ABLATION EXPERIMENTS

Our double-pessimism principle addresses two key challenges: the first component tackles the limited dataset coverage in offline RL to handle out-of-distribution issues, while the second addresses model mismatch between the data generation and deployment environments.

In this section, we conduct ablation experiments to evaluate the effectiveness of this principle. Specifically, we compare four algorithms in an offline setting: vanilla Q-learning (with zero pessimism), robust Q-learning (with model-mismatch pessimism only), offline non-robust Q-learning (with dataset pessimism only), and our proposed offline robust Q-learning (with double pessimism). The experiments are conducted on two Garnet problems, where we evaluate the robust value functions of the learned policies with respect to an uncertainty set defined by the $l_\infty$-norm.

The results are shown in Figure 6. The solid curve represents the average value across 10 independent runs, while the shaded area indicates the maximum and minimum values observed.

Our double-pessimism approach outperforms all four algorithms, including those with a single source of pessimism, demonstrating the effectiveness of our framework. Furthermore, the single-pessimism methods achieve better performance than the vanilla algorithm with no pessimism, highlighting the benefits of incorporating pessimism in offline robust RL. However, both are ultimately outperformed by our double-pessimism method, underscoring the importance of addressing both sources of uncertainty through the double-pessimism principle.

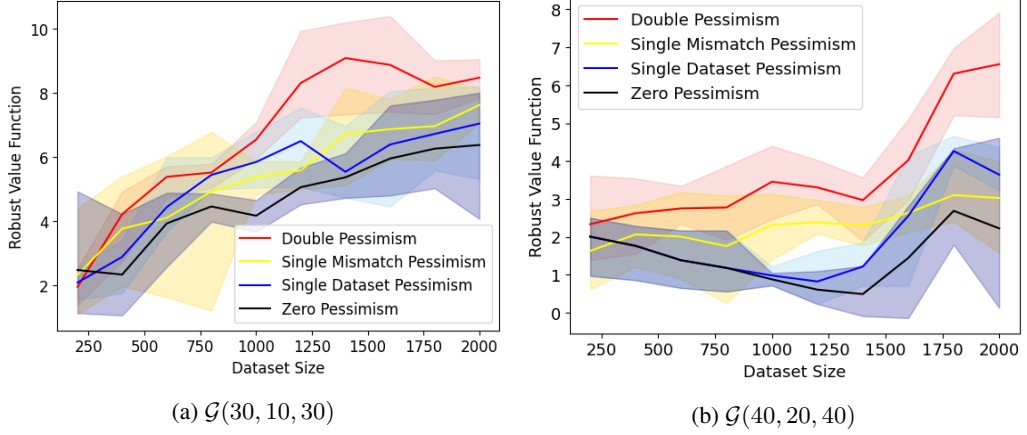

(a) $\mathcal{G}(30, 10, 30)$          (b) $\mathcal{G}(40, 20, 40)$

Figure 6: Robust value functions in Garnet problems.

## C FURTHER DISCUSSION OF $\kappa$

### C.1 A UNIVERSAL CONSTRUCTION OF $\kappa$

In this section, we discuss the design of the penalty function $\kappa$ for universal uncertainty set models defined by some distribution divergence/distance functions $F(\cdot||\cdot)$:

$$\mathcal{P} = \{P + q \in \Delta(\mathcal{S}) : F(P + q||P) \leq R\}. \tag{24}$$

Note that this uncertainty set includes perturbed environments within a region centered around the nominal kernel, effectively modeling environmental uncertainty in practical applications. This is because, in practice, perturbed environments should not deviate significantly from the nominal kernel and should therefore fall within a defined region.

We first present the following theorem for a universal construction of the penalty function $\kappa$.

**Theorem 4.** *Let $\kappa(V)$ be the optimal value of the following constrained problem:*

$$\max_q -\sum q_i V_i, \quad s.t. \ \sum_i q_i = 0, \ F(P + q||P) \leq R. \tag{25}$$

*Then, $\kappa(V)$ satisfies equation 15, i.e.,*

$$PV - \kappa(V) \leq \sigma_{\mathcal{P}}(V). \tag{26}$$

*Proof.* Note that the problem in equation 25 is equivalent to the problem

$$\max_{q \in \mathcal{Q}} -qV, \quad \text{where } \mathcal{Q} = \{q \in \mathbb{R}^S, \sum_i q_i = 0, F(P + q \| P) \leq R\}. \tag{27}$$

The proof is then straightforward by noting that $\mathcal{P} \subset \mathcal{Q}$, hence

$$PV - \kappa(V) \leq \min_{p \in \mathcal{P}} pV = \sigma_{\mathcal{P}}(V). \tag{28}$$

$\square$

Such a result provides a universal construction of the penalty function $\kappa$, for the perturbed-based uncertainty set as in equation 24. Note that $\kappa(V)$ depends on $P$, which is unknown in practice, but any unbiased estimation of it is sufficient. To illustrate this and show the generality of our design, we develop a case study for the $\chi^2$-divergence uncertainty set in the following section.

## C.2 CASE STUDY: $l_\alpha$-NORM UNCERTAINTY SET

In this section, we provide a more detailed discussion on the $l_\alpha$-norm uncertainty set. As discussed, we consider the relaxed $l_\alpha$-norm uncertainty set:

$$\tilde{\mathcal{P}}_{s,a} = \{P_{s,a} + q : \sum q_i = 0, \|q\|_\alpha \leq R_{s,a}\}, \tag{29}$$

where we relax the condition $P_{s,a} + q \geq 0$. Then the worst-case transition w.r.t. $\tilde{\mathcal{P}}$ can be derived as

$$\sigma_{\tilde{\mathcal{P}}_{s,a}}(V) = P_{s,a}V - \kappa(V), \tag{30}$$

where

$$\kappa(V) \triangleq R \min_{w \in \mathbb{R}} \|we - V\|_\beta, \tag{31}$$

with $\beta = \frac{1}{1 - \frac{1}{\alpha}}$. For popular choices of $\alpha$, the optimization problem in equation 31 has a closed-form solution, specified in Table 2 (Kumar et al., 2023). Note that for the three choices of $\alpha = 1, 2, \infty$,

| $\alpha$ | $\kappa(v)$ |
|---|---|
| $\infty$ | $\frac{\max_s v(s) - \min_s v(s)}{2}$ |
| $2$ | $\sqrt{\sum_s \left( v(s) - \frac{\sum_s v(s)}{S} \right)^2}$ |
| $1$ | $\sum_{i=1}^{\lfloor (S+1)/2 \rfloor} v(s_i) - \sum_{i=\lfloor (S+1)/2 \rfloor}^S v(s_i)$ |

Table 2: Penalty term for $l_\alpha$-norm uncertainty set

the resulting penalty terms incur a computational complexity of $\mathcal{O}(S)$. When combined with our algorithm, this leads to an overall implementation complexity of $\mathcal{O}(SA)$ per step. In contrast, the model-based methods proposed in (Shi & Chi, 2022; Blanchet et al., 2023) have a computational cost of $\mathcal{O}(S^2A)$ per step (Kumar et al., 2023), highlighting the superior efficiency and scalability of our approach.

Our algorithm and theoretical result will then characterize the convergence to the optimal robust policy w.r.t. $\tilde{\mathcal{P}}$. More importantly, when the uncertainty radius $R$ is small, the relaxation will not be effective, i.e., $\tilde{\mathcal{P}} = \mathcal{P}$ (Zhou et al., 2024).

## C.3 CASE STUDY: $\chi^2$ UNCERTAINTY SET

We adapt the construction we obtained to the widely used $\chi^2$-divergence as a case study. The design of $\kappa$ for other uncertainty sets can be obtained in a similar way.

Specifically, the uncertainty defined for the $(s, a)$-pair is

$$\mathcal{P}_{s,a} = \{P_{s,a} + q \in \Delta(\mathcal{S}) : D_{\chi^2}(P_{s,a} + q || P_{s,a}) \le R_{s,a}\}, \tag{32}$$

where $D_{\chi^2}(p||q) = \sum_i \frac{(p_i - q_i)^2}{q_i}$ is the $\chi^2$-divergence. We aim to design a model-free function $\kappa$ that serves as the penalty term to address the uncertainty from the model mismatch.

We first establish the following lemma.

**Lemma 5.** *The constrained problem*

$$\min_q \sum_i q_i V_i, \ s.t. \ \sum_i q_i = 0, D_{\chi^2}(q + P_{s,a} || P_{s,a}) \le R_{s,a} \tag{33}$$

*has the solution*

$$-\sqrt{R_{s,a} \operatorname{Var}_{P_{s,a}}(V)}. \tag{34}$$

*Proof.* To simplify the notation, we omit the subscript $s, a$ from $P_{s,a}$ and $R_{s,a}$. We note that if any entry $P_i = 0$, then any feasible $q_i = 0$, otherwise the $\chi^2$-divergence will be infinite. Thus, we can simply ignore the $i$-th entry in this case and only consider the remaining ones. Hence, we assume $P_i > 0, \forall i$ without loss of generality.

Note that the condition $D_{\chi^2}(q + P||P) \le R$ is equivalent to

$$\sum_i \frac{q_i^2}{P_i} \le R, \tag{35}$$

hence the Lagrangian function $L$ of the constrained problem is

$$L = \sum_i q_i V_i + \lambda \sum_i q_i + \mu \left(\sum_i \frac{q_i^2}{P_i} - R\right). \tag{36}$$

From the KKT conditions (Bertsekas, 1997), the solution $q^*$ and the Lagrangian multipliers $\lambda^*$ and $\mu^*$ must satisfy

$$V_i + \lambda^* + \mu^* \frac{2q_i^*}{P_i} = 0, \forall i. \tag{37}$$

We first show that if $q^*$ is the optimal solution, then $D_{\chi^2}(q^* + P||P) < R$. To show that this statement always holds, our claim is that there exists an optimal solution such that $\mu^* = 0$, then we have

$$V_i + \lambda^* = 0, \forall i \tag{38}$$

and hence,

$$\sum_i q_i^* V_i = -\lambda^* \sum_i q_i^* = 0. \tag{39}$$

To prove that this claim is not possible, we provide a counterexample to demonstrate that $\mu^* = 0$ and $\sum_i q_i^* V_i \ne 0$, which is a contradiction:

For $V = [V_1, V_2]$ and $P_{s,a} = [p_1, p_2]$, where $p_1, p_2 \ne 0$. We have $q_2 = -q_1$, then

$$\frac{q_1^2}{p_1} + \frac{q_2^2}{p_2} < R. \tag{40}$$

Hence,

$$|q_1| < \sqrt{\frac{R}{\frac{1}{p_1} + \frac{1}{p_2}}} \tag{41}$$

The optimal value of the optimization problem is

$$\sum_i q_i^* V_i = q_1^*(V_1 - V_2). \tag{42}$$

Obviously, the optimization problem does not have an optimal solution, but instead an infimum. There always exists a feasible solution $\hat{q}$ such that

$$\sum_i \hat{q}_i V_i < 0, \tag{43}$$

which means that $\mu^* \neq 0$ always holds.

Thus,

$$q_i^* V_i = -\lambda^* q_i^* - 2\mu^* \frac{(q_i^*)^2}{P_i}, \forall i, \tag{44}$$

and hence,

$$\sum_i q_i^* V_i = -2\mu^* \sum_i \frac{(q_i^*)^2}{P_i} = -2\mu^* R, \tag{45}$$

where we use the constraint $\sum_i q_i^* = 0$ and $\sum_i \frac{(q_i^*)^2}{P_i} = R$.

Again, from equation 37, we have that

$$4(\mu^*)^2 \left(\frac{q_i^*}{P_i}\right)^2 = (V_i + \lambda^*)^2, \tag{46}$$

and hence,

$$\left(\frac{q_i^*}{P_i}\right)^2 = \frac{(V_i + \lambda^*)^2}{4(\mu^*)^2}. \tag{47}$$

Taking the sum over $i$ implies that

$$\sum_i \frac{(q_i^*)^2}{P_i} = R = \sum_i P_i \frac{(V_i + \lambda^*)^2}{4(\mu^*)^2}, \tag{48}$$

and hence,

$$2\mu^* R = \sqrt{R \sum_i P_i (V_i + \lambda^*)^2}. \tag{49}$$

On the other hand, note that equation 37 further implies that

$$0 = \sum_i P_i V_i + \lambda^* \sum_i P_i, \tag{50}$$

and hence,

$$\lambda^* = -\sum_i P_i V_i. \tag{51}$$

Plugging in equation 49 implies that

$$2\mu^* R = \sqrt{R \mathrm{Var}_P(V)}. \tag{52}$$

Hence, from equation 45, the optimal solution of the constrained problem is then $-\sqrt{R \mathrm{Var}_P(V)}$, which completes the proof. $\qquad \square$

With the optimal solution to equation 33, we can then design the penalty function $\kappa$ for the $\chi^2$ uncertainty set defined as in equation 32. Firstly, we note that equation 33 is a relaxation of the support function over equation 32, therefore the optimal solution to equation 33 is not greater than

$\sigma_{\mathcal{P}}(V)$, and therefore is a pessimistic penalty of the model mismatch. We thus design the penalty function as

$$\kappa(V) = \sum_i P_i V_i - \sqrt{R \mathrm{Var}_P(V)}. \tag{53}$$

We note that in the model-free setting, it is straightforward to obtain an unbiased estimation of $\kappa$, which however requires more than 1 sample. Specifically, for $n$ i.i.d. samples $(s, a, s'_i), i = 1, ..., n$, the model-free penalty function is defined as

$$\kappa(V) = \bar{V} - \sqrt{R}\sqrt{\frac{\sum_{i=1}^n (V(s'_i) - \bar{V})^2}{n - 1}}, \tag{54}$$

where $\bar{V} = \frac{\sum_i V(s'_i)}{n}$. Such a penalty function satisfies the condition equation 15 of the pessimism principle, and hence we can extend our model-free algorithms to the $\chi^2$-divergence model. We present the algorithm for the infinite horizon in Algorithm 2. Different from Algorithm 3, for the

---

**Algorithm 2** Double-Pessimism Q-Learning for infinite-horizon RMDPs with $\chi^2$-divergence uncertainty set.

---

**Input:** $\mathcal{D}$, target success probability $1 - \delta$, $\Gamma = \left\lceil \frac{4}{1-\gamma} \log \frac{ST}{\delta} \right\rceil$

**Initialize:** $Q_0(s, a) = 0, V_0(s) = 0, n_0(s, a) = 0, \forall s, a$

**for** $t = 1, ..., T$ **do**

  Sample 2 samples $(s_{t-1}, a_{t-1}, s_t^1), (s_{t-1}, a_{t-1}, s_t^2)$ from $\mathcal{D}$

  $n_t(s_{t-1}, a_{t-1}) \leftarrow n_{t-1}(s_{t-1}, a_{t-1}) + 2; n_t(s, a) \leftarrow n_{t-1}(s, a), \forall (s, a) \neq (s_{t-1}, a_{t-1})$

  $n \leftarrow n_t(s, a); \eta_n \leftarrow (\Gamma + 1)/(\Gamma + n)$

  $b_n \leftarrow c_b \sqrt{\frac{\Gamma \log(ST/\delta)}{n(1-\gamma)^2}}$

  $M \leftarrow \frac{V_{t-1}(s_t^1) + V_{t-1}(s_t^2)}{2}$

  $\kappa \leftarrow -\sqrt{R_{s_t, a_t} \left( (V_{t-1}(s_t^1) - M)^2 + (V_{t-1}(s_t^2) - M)^2 \right)}$

  $Q_t(s_{t-1}, a_{t-1}) = (1 - \eta_n) Q_{t-1}(s_{t-1}, a_{t-1}) + \eta_n \left\{ r(s_{t-1}, a_{t-1}) + \gamma M - \gamma \kappa - b_n \right\}$

  $Q_t(s, a) = Q_{t-1}(s, a)$ for all $(s, a) \neq (s_{t-1}, a_{t-1})$

  $V_t(s_{t-1}) = \max \left\{ \max_{a \in \mathcal{A}} Q_t(s_{t-1}, a), V_{t-1}(s_{t-1}) \right\},$

  $V_t(s) = V_{t-1}(s)$ for all $s \neq s_{t-1}$.

**end for**

$\hat{\pi}(s) = \arg\max_{a \in \mathcal{A}} Q_T(s, a), \forall s$

**Output:** $\hat{\pi}$

---

$\chi^2$-divergence model, we require 2 samples at each step to estimate $\kappa$. However, the estimation does not required any information on $P_{s,a}$ and hence Algorithm 3 is still model-free.

Note that generally the penalty function $\kappa$ is biased, thus the algorithm may not converge to the optimal robust policy. However, the robustness can still be enhanced due to the additional pessimism. We validate the effectiveness of our algorithm in optimizing performance under model mismatch in an offline setting through numerical experiments. Specifically, we implemented our algorithm alongside the baseline single-pessimism Q-learning algorithm (Yan et al., 2022) on Garnet problems with varying parameters, and three simulation environments: Frozen-Lake, Taxi, and American Option. Using datasets of different sizes, we computed the robust value function of the learned policy via dynamic programming (Iyengar, 2005), and plotted the results in Figure 7 and Figure 8. Each curve represents the average over 10 independent runs, with the shaded region indicating the maximum and minimum values. As demonstrated in the results, our double-pessimism Q-learning significantly outperforms the single-pessimism approach, showcasing the robustness of our algorithm to model mismatch and confirming the efficacy of our double-pessimism design.

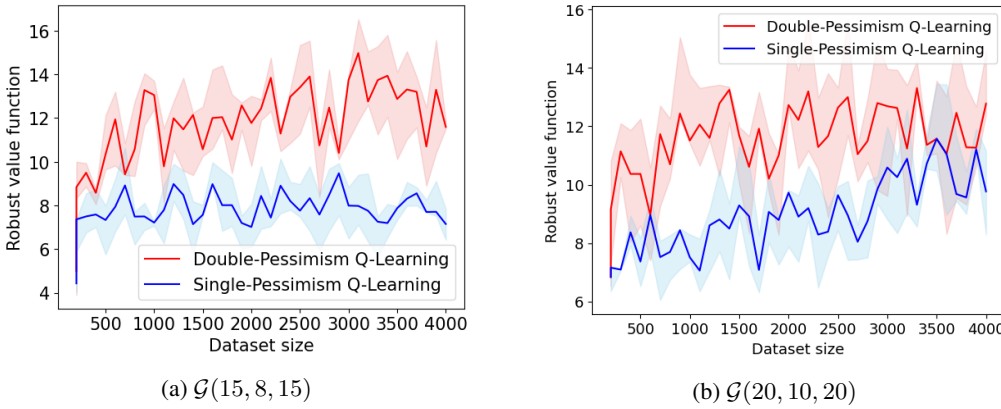

(a) $\mathcal{G}(15, 8, 15)$          (b) $\mathcal{G}(20, 10, 20)$

Figure 7: Robust value functions of two Granet problems over $\chi^2$-divergence uncertainty set. Solid lines represent the mean values over 10 independent runs. Shaded areas represent the maximum and minimum values.

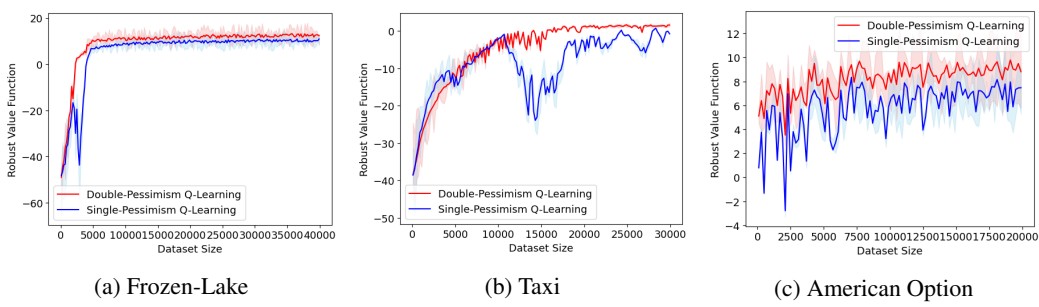

(a) Frozen-Lake         (b) Taxi         (c) American Option

Figure 8: Robust value functions of three simulation environments over the $\chi^2$-divergence uncertainty set. Solid lines represent the mean values over 10 independent runs. Shaded areas represent the maximum and minimum values.

## D    ANALYSIS OF THE FINITE HORIZON SETTING

### D.1    NOTATION

Recall the learning rate defined by

$$\eta_n = \frac{H+1}{H+n} \tag{55}$$

for the $n$-th visit of a given state-action pair at a given time step $h$. We further adopt two sequences of related quantities for any integers $N \geq 0$ and $n \geq 1$ from (Shi et al., 2022):

$$\eta_0^N \triangleq \begin{cases} \prod_{i=1}^N (1-\eta_i) = 0, & \text{if } N > 0, \\ 1, & \text{if } N = 0 \end{cases} , \tag{56}$$

$$\eta_n^N \triangleq \begin{cases} \eta_n \prod_{i=n+1}^N (1-\eta_i), & \text{if } N > n, \\ \eta_n, & \text{if } N = n, \\ 0, & \text{if } N < n \end{cases} . \tag{57}$$

It has been shown in (Shi et al., 2022; Yan et al., 2022) that

$$\sum_{n=0}^N \eta_n^N = 1. \tag{58}$$

We also introduce the following notation:

- $N_h^k(s, a)$, or simply $N_h^k$: The number of episodes that have visited the state-action pair $(s, a)$ at step $h$ before the start of the $k$-th episode.

- $k_h^n(s, a)$, or simply $k^n$: The index of the episode in which the state-action pair $(s, a)$ is visited at step $h$ for the $n$-th time. We adopt the convention that $k^0 = 0$.

- $P_h^k \in \{0, 1\}^{1 \times S}$: A row vector corresponding to the empirical transition at step $h$ of the $k$-th episode, defined as

$$P_h^k(s) = \mathbf{1}\big(s = s_{h+1}^k\big) \qquad \text{for all } s \in \mathcal{S}. \tag{59}$$

- $\pi^k = \{\pi_h^k\}_{h=1}^H$ with $\pi_h^k(s) \triangleq \arg\max_a Q_h^k(s, a)$ for all $(h, s) \in [H] \times \mathcal{S}$: The deterministic greedy policy at the beginning of the $k$-th episode.

- $\widehat{\pi}$: The final output of the algorithm, corresponding to $\pi^{K+1}$ as defined above. For simplicity in our analysis, we treat $\widehat{\pi}$ as $\pi^K$, which does not affect the result.

### D.2 LEMMAS FOR THEOREM 2

In this section, we present the lemmas that are utilized in the proof of Theorem 2.

The first lemma demonstrates how our choice of the penalty term $\kappa$ can address the uncertainty arising from model mismatch.

**Lemma 6.** *(Theorem 1 in (Kumar et al., 2023)) Let $\mathcal{P}_{s,a}$ be the uncertainty set defined using the $l_\alpha$-norm. For any vector $V$, the following relationship holds:*

$$\sigma_{\mathcal{P}_{s,a}}(V) = P_{s,a}V - \kappa_{s,a}(V), \tag{60}$$

*where $\kappa$ is defined as in equation 18.*

The following lemma provides properties concerning the learning rates and is adapted from (Jin et al., 2018; Li et al., 2021).

**Lemma 7** (Lemma 1 in (Li et al., 2021)). *For any integer $N > 0$, the following properties hold:*

$$\frac{1}{N^a} \leq \sum_{n=1}^N \frac{\eta_n^N}{n^a} \leq \frac{2}{N^a} \qquad \text{for all} \quad \frac{1}{2} \leq a \leq 1, \tag{61a}$$

$$\max_{1 \leq n \leq N} \eta_n^N \leq \frac{2H}{N}, \qquad \sum_{n=1}^N (\eta_n^N)^2 \leq \frac{2H}{N}, \qquad \sum_{n=N}^\infty \eta_n^N \leq 1 + \frac{1}{H}. \tag{61b}$$

The following lemmas concern the concentration properties of the sample generation.

The first lemma below is adapted from Xie et al. (2021b, Lemma A.1).

**Lemma 8.** *(Lemma 8 in (Shi et al., 2022)) Suppose $N \sim \mathsf{Binomial}(n, p)$, where $n \geq 1$ and $p \in [0, 1]$. For any $\delta \in (0, 1)$, we have*

$$\frac{p}{N \vee 1} \leq \frac{8 \log\left(\frac{1}{\delta}\right)}{n}, \tag{62}$$

*and*

$$N \geq \frac{np}{8 \log\left(\frac{1}{\delta}\right)} \qquad \text{if } np \geq 8 \log\left(\frac{1}{\delta}\right), \tag{63a}$$

$$N \leq \begin{cases} e^2 np & \text{if } np \geq \log\left(\frac{1}{\delta}\right), \\ 2e^2 \log\left(\frac{1}{\delta}\right) & \text{if } np \leq 2 \log\left(\frac{1}{\delta}\right). \end{cases} \tag{63b}$$

*with probability at least $1 - 4\delta$.*

The following lemma is a standard concentration inequity result.

**Theorem 9** (Freedman's inequality (Freedman, 1975)). *Consider a filtration $\mathcal{F}_0 \subset \mathcal{F}_1 \subset \mathcal{F}_2 \subset \cdots$, and let $\mathbb{E}_k$ stand for the expectation conditioned on $\mathcal{F}_k$. Suppose that $Y_n = \sum_{k=1}^n X_k \in \mathbb{R}$, where $\{X_k\}$ is a real-valued scalar sequence obeying*

$$|X_k| \leq R \qquad \text{and} \qquad \mathbb{E}_{k-1}[X_k] = 0 \qquad \text{for all } k \geq 1$$

*for some quantity $R < \infty$. We also define*

$$W_n := \sum_{k=1}^n \mathbb{E}_{k-1}[X_k^2].$$

*In addition, suppose that $W_n \leq \sigma^2$ holds deterministically for some given quantity $\sigma^2 < \infty$. Then, for any positive integer $m \geq 1$, with probability at least $1 - \delta$ one has*

$$|Y_n| \leq \sqrt{8 \max\left\{W_n, \frac{\sigma^2}{2^m}\right\} \log \frac{2m}{\delta}} + \frac{4}{3} R \log \frac{2m}{\delta}. \tag{64}$$

The Freedman's inequality further implies several important results related to our problem.

**Lemma 10.** *Let $\{W_h^i \in \mathbb{R}^S \mid 1 \leq i \leq K, 1 \leq h \leq H+1\}$ be a collection of vectors satisfying the following properties:*

- *$W_h^i$ is fully determined by the samples collected up to the end of the $(h-1)$-th step of the $i$-th episode;*

- *$\|W_h^i\|_\infty \leq C_{\mathrm{w}}$.*

*For any positive integer $N \geq H$, consider the following sequence:*

$$X_i(s, a, h, N) \triangleq \eta_{N_h^i(s,a)}^N \left(P_h^i W_{h+1}^i - R_{s,a} \kappa(W_{h+1}^i) - \sigma_{h,s,a}(W_{h+1}^i)\right) \boldsymbol{I}\{(s_h^i, a_h^i) = (s, a)\}. \tag{65}$$

*With probability at least $1 - \delta$,*

$$\left| \sum_{i=1}^k X_i(s, a, h, N) \right| \lesssim \sqrt{\frac{H}{N} C_{\mathrm{w}}^2 \log^2 \frac{SAT}{\delta}} \tag{66}$$

*holds simultaneously for all $(k, h, s, a, N) \in [K] \times [H] \times \mathcal{S} \times \mathcal{A} \times [K]$.*

*Proof.* Let $u_h^i(s, a, N) = \eta_{N_h^i(s,a)}^N$. From equation 61b in Lemma 7, we have

$$\left| u_h^i(s, a, N) \right| \leq \frac{2H}{N} \triangleq C_{\mathrm{u}}.$$

Given that $\mathrm{Var}_{h,s,a}\left(W_{h+1}^{k_h^n(s,a)}\right) \leq C_{\mathrm{w}}^2$, we can apply Lemma 7 from (Li et al., 2021) to obtain, with probability at least $1 - \delta$,

$$\left| \sum_{i=1}^k X_i(s, a, h, N) \right|$$

$$\lesssim \sqrt{C_{\mathrm{u}} \log^2 \frac{SAT}{\delta}} \sqrt{\sum_{n=1}^{N_h^k(s,a)} \eta_n^N C_{\mathrm{w}}^2} + \left(C_{\mathrm{u}} C_{\mathrm{w}} + \sqrt{\frac{C_{\mathrm{u}}}{N}} C_{\mathrm{w}}\right) \log^2 \frac{SAT}{\delta}$$

$$\lesssim \sqrt{\frac{H}{N} \log^2 \frac{SAT}{\delta}} \cdot C_{\mathrm{w}} + \frac{H C_{\mathrm{w}}}{N} \log^2 \frac{SAT}{\delta}$$

$$\lesssim \sqrt{\frac{H C_{\mathrm{w}}^2}{N} \log^2 \frac{SAT}{\delta}},$$

where the final line uses equation 61b from Lemma 7 again. $\qquad\square$

**Lemma 11.** *Let $\left\{ W_h^k(s,a) \in \mathbb{R}^S \mid (s,a) \in \mathcal{S} \times \mathcal{A}, 1 \le k \le K, 1 \le h \le H+1 \right\}$ be a collection of vectors satisfying the following properties:*

- *$W_h^k(s,a)$ is fully determined by the given state-action pair $(s,a)$ and the samples collected up to the end of the $(k-1)$-th episode;*

- *$\|W_h^k(s,a)\|_\infty \le C_{\mathrm{w}}$.*

*For any positive $C_{\mathrm{d}} \ge 0$, consider the following sequences:*

$$X_{h,k} \triangleq C_{\mathrm{d}} \left[ \frac{d_{P,h}^{\pi_\star}(s_h^k, a_h^k)}{d_{P,h}^\mu(s_h^k, a_h^k)} W_{h+1}^k(s_h^k, a_h^k) - \sum_{(s,a) \in \mathcal{S} \times \mathcal{A}} d_{P,h}^{\pi_\star}(s,a) W_{h+1}^k(s,a) \right], \qquad (67)$$

$$\overline{X}_{h,k} \triangleq C_{\mathrm{d}} \left[ \frac{d_{P,h}^{\pi_\star}(s_h^k, a_h^k)}{d_{P,h}^\mu(s_h^k, a_h^k)} W_{h+1}^k(s_h^k, a_h^k) - \sum_{(s,a) \in \mathcal{S} \times \mathcal{A}} d_{P,h}^{\pi_\star}(s,a) W_{h+1}^k(s,a) \right]. \qquad (68)$$

*Consider any $\delta \in (0,1)$. Then with probability at least $1 - \delta$,*

$$\left| \sum_{k=1}^K X_{h,k} \right| \le \sqrt{\sum_{k=1}^K 8 C_{\mathrm{d}}^2 C^\star \sum_{(s,a) \in \mathcal{S} \times \mathcal{A}} d_{P,h}^{\pi_\star}(s,a) \left[ P_{h,s,a} W_{h+1}^k(s,a) \right]^2 \log \frac{2H}{\delta}} + 2 C_{\mathrm{d}} C^\star C_{\mathrm{w}} \log \frac{2H}{\delta},$$
$$\qquad (69)$$

$$\left| \sum_{k=1}^K \overline{X}_{h,k} \right| \le \sqrt{\sum_{k=1}^K 8 C_{\mathrm{d}}^2 C^\star \sum_{(s,a) \in \mathcal{S} \times \mathcal{A}} d_{P,h}^{\pi_\star}(s,a) P_{h,s,a} \left[ W_{h+1}^k(s,a) \right]^2 \log \frac{2H}{\delta}} + 2 C_{\mathrm{d}} C^\star C_{\mathrm{w}} \log \frac{2H}{\delta},$$
$$\qquad (70)$$

*hold simultaneously for all $h \in [H]$.*

*Proof.* The proof similarly follows from (Shi et al., 2022). $\qquad \square$

We then prove Lemma 1 showing the effectiveness of our double pessimism principle, i.e., that our estimation is a conservative estimation of the robust value function.

**Lemma 12.** *Consider any $\delta \in (0,1)$, and suppose that $c_b > 0$ is some sufficiently large constant. Then, with probability at least $1 - \delta$,*

$$\left| \sum_{n=1}^{N_h^k(s,a)} \eta_n^{N_h^k(s,a)} \left( \sigma_{h,s,a}(V_{h+1}^{k^n(s,a)}) - P_h^{k^n(s,a)} V_{h+1}^{k^n(s,a)} + R_{s,a} \kappa(V_{h+1}^{k^n(s,a)}) \right) \right| \le \sum_{n=1}^{N_h^k(s,a)} \eta_n^{N_h^k(s,a)} b_n$$
$$\qquad (71)$$

*holds simultaneously for all $(k,h,s,a) \in [K] \times [H] \times \mathcal{S} \times \mathcal{A}$, and*

$$V_h^k(s) \le V_h^{\pi^k}(s) \le V_h^\star(s) \qquad (72)$$

*holds simultaneously for all $(k,h,s) \in [K] \times [H] \times \mathcal{S}$.*

*Proof.* **Proof of inequality equation 71.** We show it by invoking Lemma 10. Let

$$W_{h+1}^i := V_{h+1}^i,$$

which satisfies

$$\|W_{h+1}^i\|_\infty \le H =: C_{\mathrm{w}}.$$

Note that it holds that

$$\sigma_{h,s,a}(V_{h+1}^{k^n(s,a)}) - P_h^{k^n(s,a)} V_{h+1}^{k^n(s,a)} + R_{s,a} \kappa(V_{h+1}^{k^n(s,a)})$$
$$= P_{h,s,a} V_{h+1}^{k^n(s,a)} - R_{s,a} \kappa(V_{h+1}^{k^n(s,a)}) - P_h^{k^n(s,a)} V_{h+1}^{k^n(s,a)} + R_{s,a} \kappa(V_{h+1}^{k^n(s,a)})$$

$$= P_{h,s,a} V_{h+1}^{k^n(s,a)} - P_h^{k^n(s,a)} V_{h+1}^{k^n(s,a)}, \tag{73}$$

where the first equation is from Lemma 6. Hence applying Lemma 10 implies that with probability at least $1 - \delta$,

$$\sum_{n=1}^{N_h^k(s,a)} \eta_n^{N_h^k(s,a)} \Big( \sigma_{h,s,a}(V_{h+1}^{k^n(s,a)}) - P_h^{k^n(s,a)} V_{h+1}^{k^n(s,a)} + R_{s,a} \kappa(V_{h+1}^{k^n(s,a)}) \Big)$$

$$= \left| \sum_{n=1}^{N_h^k(s,a)} \eta_n^{N_h^k(s,a)} \Big( P_{h,s,a} - P_h^{k^n(s,a)} \Big) V_{h+1}^{k^n(s,a)} \right|$$

$$= \left| \sum_{i=1}^{k} X_i\big(s,a,h,N_h^k(s,a)\big) \right|$$

$$\le c_b \sqrt{\frac{H^3 \iota^2}{N_h^k(s,a)}} \tag{74}$$

holds simultaneously for all $(s,a,k,h) \in \mathcal{S} \times \mathcal{A} \times [K] \times [H]$, provided that the constant $c_b > 0$ is large enough and that $N = N_h^k(s,a) > 0$. When $N_h^k(s,a) = 0$, we have the trivial bound

$$\left| \sum_{n=1}^{N_h^k(s,a)} \eta_n^{N_h^k(s,a)} \Big( P_{h,s,a} - P_h^{k^n(s,a)} \Big) V_{h+1}^{k^n(s,a)} \right| = 0. \tag{75}$$

Additionally, from the definition $b_n = c_b \sqrt{\frac{H^3 \iota^2}{n}}$, we observe that

$$\begin{cases} \sum_{n=1}^{N_h^k(s,a)} \eta_n^{N_h^k(s,a)} b_n \in \left[ c_b \sqrt{\frac{H^3 \iota^2}{N_h^k(s,a)}}, 2 c_b \sqrt{\frac{H^3 \iota^2}{N_h^k(s,a)}} \right], & \text{if } N_h^k(s,a) > 0 \\ \sum_{n=1}^{N_h^k(s,a)} \eta_n^{N_h^k(s,a)} b_n = 0, & \text{if } N_h^k(s,a) = 0 \end{cases} \tag{76}$$

holds simultaneously for all $s,a,h,k \in \mathcal{S} \times \mathcal{A} \times [H] \times [K]$, which follows directly from the property equation 61a in Lemma 7.

Combining the above, equation 74 and equation 76 hence imply that

$$\left| \sum_{n=1}^{N_h^k(s,a)} \eta_n^{N_h^k(s,a)} \Big( \sigma_{h,s,a}(V_{h+1}^{k^n(s,a)}) - P_h^{k^n(s,a)} V_{h+1}^{k^n(s,a)} + R_{s,a} \kappa(V_{h+1}^{k^n(s,a)}) \Big) \right|$$

$$\le \sum_{n=1}^{N_h^k(s,a)} \eta_n^{N_h^k(s,a)} b_n.$$

**Proof of inequality equation 72.** Note that the second inequality of equation 72 is straightforward as

$$V_h^\pi(s) \le V^\star(s)$$

holds for any policy $\pi$. As a consequence, it suffices to establish the first inequality of equation 72:

$$V_h^k(s) \le V_h^{\pi^k}(s) \qquad \text{for all } (s,h,k) \in \mathcal{S} \times [H] \times [K]. \tag{77}$$

Define

$$k_o(h,k,s) := \max \left\{ l : l < k \text{ and } V_h^l(s) = \max_a Q_h^l(s,a) \right\} \tag{78}$$

for any $(h,k,s) \in [H] \times [K] \times \mathcal{S}$, which denotes the index of the latest episode — before the end of the $(k-1)$-th episode — in which $V_h(s)$ has been updated. We abbreviate $k_o(h,k,s)$ as $k_o(h)$ whenever it is clear from the context.

We utilize an induction approach to show that. Assume that

$$V_\Gamma^{k'}(s) \le V_\Gamma^{\pi^{k'}}(s) \qquad \text{for all } (k',\Gamma,s) \in [k-1] \times [H+1] \times \mathcal{S}, \tag{79a}$$

$$V_\Gamma^k(s) \le V_\Gamma^{\pi^k}(s) \qquad \text{for all } \Gamma \ge h+1 \text{ and } s \in \mathcal{S}. \tag{79b}$$

We need to verify

$$V_h^k(s) \le V_h^{\pi^k}(s) \qquad \text{for all } s \in \mathcal{S}. \tag{80}$$

*Step 1: base case.*

Let us begin with the base case when $h+1 = H+1$ for all episodes $k \in [K]$. Recognizing the fact that $V_{H+1}^\pi = V_{H+1}^k = 0$ for any $\pi$ and any $k \in [K]$, we directly arrive at

$$V_{H+1}^k(s) \le V_{H+1}^{\pi^k}(s) \qquad \text{for all } (k,s) \in [K] \times \mathcal{S}. \tag{81}$$

*Step 2: induction.* To justify equation 80 under the induction hypothesis equation 79, we decompose the difference term to obtain

$$V_h^{\pi^k}(s) - V_h^k(s) = V_h^{\pi^k}(s) - \max\left\{\max_a Q_h^k(s,a), V_h^{k-1}(s)\right\}$$
$$= Q_h^{\pi^k}\left(s, \pi_h^k(s)\right) - \max\left\{\max_a Q_h^k(s,a), V_h^{k_o(h)}(s)\right\}, \tag{82}$$

where the last line holds since $V_h(s)$ has not been updated during episodes $k_o(h), k_o(h)+1, \cdots, k-1$ (in view of the definition of $k_o(h)$ in equation 78). We shall prove that the right-hand side of equation 82 is non-negative by discussing the following two cases separately.

**Case 1.** Consider the case where $V_h^k(s) = \max_a Q_h^k(s,a)$. Note that

$$\pi_h^k(s) = \arg\max_a Q_h^k(s,a), \qquad \text{when } V_h^k(s) = \max_a Q_h^k(s,a) \tag{83}$$

holds for all $(k,h) \in [K] \times [H]$, Thus

$$V_h^{\pi^k}(s) - V_h^k(s) = Q_h^{\pi^k}\left(s, \pi_h^k(s)\right) - \max_a Q_h^k(s,a)$$
$$= Q_h^{\pi^k}\left(s, \pi_h^k(s)\right) - Q_h^k\left(s, \pi_h^k(s)\right). \tag{84}$$

To continue, we turn to controlling a more general term $Q_h^{\pi^k}(s,a) - Q_h^k(s,a)$ for all $(s,a) \in \mathcal{S} \times \mathcal{A}$. Invoking the fact $\eta_0^{N_h^k} + \sum_{n=1}^{N_h^k} \eta_n^{N_h^k} = 1$ (see equation 56 and equation 58) leads to

$$Q_h^{\pi^k}(s,a) = \eta_0^{N_h^k} Q_h^{\pi^k}(s,a) + \sum_{n=1}^{N_h^k} \eta_n^{N_h^k} Q_h^{\pi^k}(s,a).$$

This relation combined with equation 106 allows us to express the difference between $Q_h^{\pi^k}$ and $Q_h^k$ as follows

$$Q_h^{\pi^k}(s,a) - Q_h^k(s,a)$$

$$= \eta_0^{N_h^k}\left(Q_h^{\pi^k}(s,a) - Q_h^1(s,a)\right) + \sum_{n=1}^{N_h^k} \eta_n^{N_h^k}\left[Q_h^{\pi^k}(s,a) - r_h(s,a) - V_{h+1}^{k^n}(s_{h+1}^{k^n}) + R_{s,a}\kappa(V_{h+1}^{k^n}) + b_n\right]$$

$$\overset{(a)}{=} \eta_0^{N_h^k}\left(Q_h^{\pi^k}(s,a) - Q_h^1(s,a)\right) + \sum_{n=1}^{N_h^k} \eta_n^{N_h^k}\left[P_{h,s,a}V_{h+1}^{\pi^k} - R_{s,a}\kappa(V_{h+1}^{\pi^k}) - V_{h+1}^{k^n}(s_{h+1}^{k^n}) + R_{s,a}\kappa(V_{h+1}^{k^n}) + b_n\right]$$

$$\overset{(b)}{\ge} \sum_{n=1}^{N_h^k} \eta_n^{N_h^k}\left[P_{h,s,a}V_{h+1}^{\pi^k} - R_{s,a}\kappa(V_{h+1}^{\pi^k}) - V_{h+1}^{k^n}(s_{h+1}^{k^n}) + R_{s,a}\kappa(V_{h+1}^{k^n}) + b_n\right]$$

$$\overset{(c)}{=} \sum_{n=1}^{N_h^k} \eta_n^{N_h^k}\left[\sigma_{h,s,a}(V_{h+1}^{\pi^k}) - \sigma_{h,s,a}(V_{h+1}^{k^n}) + \sigma_{h,s,a}(V_{h+1}^{k^n}) - V_{h+1}^{k^n}(s_{h+1}^{k^n}) + R_{s,a}\kappa(V_{h+1}^{k^n}) + b_n\right]$$

$$\overset{(d)}{\ge} \sum_{n=1}^{N_h^k} \eta_n^{N_h^k}\left[\left(P_{h,s,a} - P_h^{k^n}\right)V_{h+1}^{k^n} + b_n\right]. \tag{85}$$

Here, (a) invokes the robust Bellman equation $Q_h^{\pi^k}(s,a) = r_h(s,a) + \sigma_{h,s,a}(V_{h+1}^{\pi^k})$; (b) holds since $Q_h^{\pi^k}(s,a) \geq 0 = Q_h^1(s,a)$; (c) is from Lemma 6; and (d) comes from the fact

$$V_{h+1}^{\pi^k} \geq V_{h+1}^k \geq V_{h+1}^{k^n},$$

owing to the induction hypothesis in equation 79 as well as the monotonicity of $V_{h+1}$ in Lemma 12. Consequently, it follows from equation 85 that

$$
\begin{aligned}
& Q_h^{\pi^k}(s,a) - Q_h^k(s,a) \\
&\geq \sum_{n=1}^{N_h^k(s,a)} \eta_n^{N_h^k(s,a)} \left( P_{h,s,a} - P_h^{k^n(s,a)} \right) V_{h+1}^{k^n(s,a)} + \sum_{n=1}^{N_h^k(s,a)} \eta_n^{N_h^k(s,a)} b_n \\
&\geq \sum_{n=1}^{N_h^k(s,a)} \eta_n^{N_h^k(s,a)} b_n - \left| \sum_{n=1}^{N_h^k(s,a)} \eta_n^{N_h^k(s,a)} \left( P_{h,s,a} - P_h^{k^n(s,a)} \right) V_{h+1}^{k^n(s,a)} \right| \\
&\geq 0
\end{aligned}
\tag{86}
$$

for all state-action pair $(s,a)$, where the last inequality holds due to the bound in equation 71 in Lemma 12. Plugging the above result into equation 84 directly establishes that

$$V_h^{\pi^k}(s) - V_h^k(s) = Q_h^{\pi^k}\big(s,\pi^k(s)\big) - Q_h^k\big(s,\pi^k(s)\big) \geq 0. \tag{87}$$

**Case 2.** When $V_h^k(s) = V_h^{k_o(h)}(s)$, it indicates that

$$V_h^{k_o(h)}(s) = \max_a Q_h^{k_o(h)}(s,a), \qquad \pi_h^{k_o(h)}(s) = \arg\max_a Q_h^{k_o(h)}(s,a), \tag{88}$$

which follows from the definition of $k_o(h)$ in equation 78 and the corresponding fact in equation 83. We also make note of the fact that

$$\pi_h^k(s) = \pi_h^{k_o(h)}(s), \tag{89}$$

which holds since $V_h(s)$ (and hence $\pi_h(s)$) has not been updated during episodes $k_o(h), k_o(h)+1, \cdots, k-1$ (in view of the definition equation 78). Combining the above two results, we can show that

$$
\begin{aligned}
V_h^{\pi^k}(s) - V_h^k(s) &= Q_h^{\pi^k}\big(s,\pi_h^k(s)\big) - V_h^{k_o(h)}(s) = Q_h^{\pi^k}\big(s,\pi_h^k(s)\big) - \max_a Q_h^{k_o(h)}(s,a) \\
&= Q_h^{\pi^k}\big(s,\pi_h^{k_o(h)}(s)\big) - Q_h^{k_o(h)}\big(s,\pi_h^{k_o(h)}(s)\big) \\
&\geq 0,
\end{aligned}
\tag{90}
$$

where the final line can be verified using exactly the same argument as in the previous case to show equation 85 and then equation 87. Here, we omit the proof of this step for brevity.

To conclude, substituting the relations equation 87 and equation 90 in the above two cases back into equation 82, we arrive at

$$V_h^{\pi^k}(s) - V_h^k(s) \geq 0$$

as desired in equation 80. This immediately completes the induction argument. $\qquad\square$

**Lemma 13.** *With probability at least $1 - \delta$, it holds that*

$$\sum_{k=1}^K \sum_{(s,a)\in\mathcal{S}\times\mathcal{A}} d_{P,h}^{\pi^\star}(s,a) \sum_{n=1}^{N_h^k(s,a)} \eta_n^{N_h^k(s,a)} \big( \sigma_{h,s,a}(V_{h+1}^\star) - \sigma_{h,s,a}(V_{h+1}^{k^n(s,a)}) \big) \tag{91}$$

$$\leq \left( 1 + \frac{1}{H} \right) \sum_{k=1}^K \sum_{s\in\mathcal{S}} d_{P,h+1}^{\pi^\star}(s) \big( V_{h+1}^\star(s) - V_{h+1}^k(s) \big) + 24\sqrt{H^2 C^\star K \log\frac{2H}{\delta}} + 12 H C^\star \log\frac{2H}{\delta}.$$

*Proof.* It is sufficient to show that

$$A_h \triangleq \sum_{k=1}^K \underbrace{\sum_{(s,a)\in\mathcal{S}\times\mathcal{A}} d_{P,h}^{\pi^\star}(s,a) \sum_{n=1}^{N_h^k(s,a)} \eta_n^{N_h^k(s,a)} \big( \sigma_{h,s,a}(V_{h+1}^\star) - \sigma_{h,s,a}(V_{h+1}^{k^n(s,a)}) \big)}_{=:A_{h,k}} \tag{92}$$

$$\leq \sum_{k=1}^{K} \left(1 + \frac{1}{H}\right) \underbrace{\sum_{s \in \mathcal{S}} d_{P,h+1}^{\pi^\star}(s) \left(V_{h+1}^\star(s) - V_{h+1}^k(s)\right)}_{=:B_{h,k}} + 24\sqrt{H^2 C^\star K \log \frac{2H}{\delta}} + 12 H C^\star \log \frac{2H}{\delta}.$$

Define two auxiliary sequences $\{Y_{h,k}\}_{k=1}^K$ and $\{Z_{h,k}\}_{k=1}^K$ which are the empirical estimates of $A_{h,k}$ and $B_{h,k}$, respectively. For any time step $h$ in episode $k$, $Y_{h,k}$ and $Z_{h,k}$ are defined as follows

$$Y_{h,k} := \frac{d_{P,h}^{\pi^\star}(s_h^k, a_h^k)}{d_{P,h}^{\mu}(s_h^k, a_h^k)} \sum_{n=1}^{N_h^k(s_h^k, a_h^k)} \eta_n^{N_h^k(s_h^k, a_h^k)} \left(\sigma_{h,s_h^k,a_h^k}(V_{h+1}^\star) - \sigma_{h,s_h^k,a_h^k}(V_{h+1}^{k^n(s_h^k, a_h^k)})\right),$$

$$Z_{h,k} := \left(1 + \frac{1}{H}\right) \frac{d_{P,h}^{\pi^\star}(s_h^k, a_h^k)}{d_{P,h}^{\mu}(s_h^k, a_h^k)} \left(\sigma_{h,s_h^k,a_h^k}(V_{h+1}^\star) - \sigma_{h,s_h^k,a_h^k}(V_{h+1}^k)\right).$$

Note that

$$\sum_{k=1}^K Y_{h,k} = \sum_{k=1}^K \frac{d_{P,h}^{\pi^\star}(s_h^k, a_h^k)}{d_{P,h}^{\mu}(s_h^k, a_h^k)} \sum_{n=1}^{N_h^k(s_h^k, a_h^k)} \eta_n^{N_h^k(s_h^k, a_h^k)} \left(\sigma_{h,s_h^k,a_h^k}(V_{h+1}^\star) - \sigma_{h,s_h^k,a_h^k}(V_{h+1}^{k^n(s_h^k, a_h^k)})\right)$$

$$\overset{\text{(i)}}{=} \sum_{l=1}^K \frac{d_{P,h}^{\pi^\star}(s_h^l, a_h^l)}{d_{P,h}^{\mu}(s_h^l, a_h^l)} \left\{ \sum_{N=N_h^l(s_h^l, a_h^l)}^{N_h^K(s_h^l, a_h^l)} \eta_{N_h^l(s_h^l, a_h^l)}^N \right\} \left(\sigma_{h,s_h^l,a_h^l}(V_{h+1}^\star) - \sigma_{h,s_h^l,a_h^l}(V_{h+1}^l)\right) \tag{93}$$

$$\leq \left(1 + \frac{1}{H}\right) \sum_{k=1}^K \frac{d_{P,h}^{\pi^\star}(s_h^k, a_h^k)}{d_{P,h}^{\mu}(s_h^k, a_h^k)} \left(\sigma_{h,s_h^k,a_h^k}(V_{h+1}^\star) - \sigma_{h,s_h^k,a_h^k}(V_{h+1}^k)\right) = \sum_{k=1}^K Z_{h,k}. \tag{94}$$

Here, (a) holds by replacing $k^n(s_h^k, a_h^k)$ with $l$ and gathering all terms that involve $V_{h+1}^\star - V_{h+1}^l$; in the last line, we have invoked the property $\sum_{N=n}^{N_h^K(s,a)} \eta_n^N \leq \sum_{N=n}^\infty \eta_n^N = 1 + 1/H$ (see equation 61b) together with the fact $V_{h+1}^\star - V_{h+1}^l \geq 0$ (see Lemma 12), and have further replaced $l$ with $k$.

With the above relation in hand, in order to verify equation 93, we further decompose $A_h$ into several terms

$$A_h = \sum_{k=1}^K A_{h,k} = \sum_{k=1}^K Y_{h,k} + \sum_{k=1}^K (A_{h,k} - Y_{h,k}) \overset{\text{(a)}}{\leq} \sum_{k=1}^K Z_{h,k} + \sum_{k=1}^K (A_{h,k} - Y_{h,k})$$

$$= \sum_{k=1}^K B_{h,k} + \sum_{k=1}^K (Z_{h,k} - B_{h,k}) + \sum_{k=1}^K (A_{h,k} - Y_{h,k}) \tag{95}$$

where (a) follows from equation 94.

As a result, it remains to control $\sum_{k=1}^K (Z_{h,k} - B_{h,k})$ and $\sum_{k=1}^K (A_{h,k} - Y_{h,k})$ separately in the following.

*Step 1: controlling* $\sum_{k=1}^K (A_{h,k} - Y_{h,k})$. We shall first control this term by means of Lemma 11. Specifically, consider

$$W_{h+1}^k(s,a) := \sum_{n=1}^{N_h^k(s,a)} \eta_n^{N_h^k(s,a)} \left(\sigma_{h,s,a}(V_{h+1}^\star) - \sigma_{h,s,a}(V_{h+1}^{k^n(s,a)})\right), \qquad C_{\mathrm{d}} := 1 \tag{96}$$

which satisfies

$$\left\|W_{h+1}^k(s,a)\right\|_\infty \leq \sum_{n=1}^{N_h^k(s,a)} \eta_n^{N_h^k(s,a)} \left(\left\|V_{h+1}^\star\right\|_\infty + \left\|V_{h+1}^{k^n(s,a)}\right\|_\infty\right) \leq 2H =: C_{\mathrm{w}}. \tag{97}$$

Here, we use the fact that $\eta_0^{N_h^k} + \sum_{n=1}^{N_h^k} \eta_n^{N_h^k} = 1$ (see equation 56 and equation 58). Then, applying Lemma 11 with equation 96, we have with probability at least $1 - \delta$, the following inequality holds true

$$\left| \sum_{k=1}^{K} \left( A_{h,k} - Y_{h,k} \right) \right| =: \left| \sum_{k=1}^{K} X_{h,k} \right|$$

$$\leq \sqrt{\sum_{k=1}^{K} 8C_{\mathrm{d}}^2 C^\star \sum_{(s,a) \in \mathcal{S} \times \mathcal{A}} d_{P,h}^{\pi^\star}(s,a) \left[ W_{h+1}^k(s,a) \right]^2 \log \frac{2H}{\delta} + 2C_{\mathrm{d}} C^\star C_{\mathrm{w}} \log \frac{2H}{\delta}}$$

$$\leq 16\sqrt{H^2 C^\star K \log \frac{2H}{\delta}} + 4HC^\star \log \frac{2H}{\delta}, \tag{98}$$

where the last inequality is from $\left| W_{h+1}^k(s,a) \right| \leq \left\| V_{h+1}^* - V_{h+1}^{k^n(s,a)} \right\|_\infty \leq H$.

*Step 2: controlling* $\sum_{k=1}^{K} \left( Z_{h,k} - B_{h,k} \right)$. Similarly, we shall control $\sum_{k=1}^{K} \left( Z_{h,k} - B_{h,k} \right)$ by invoking Lemma 11.

Recall that

$$Z_{h,k} - B_{h,k} = \left( 1 + \frac{1}{H} \right) \frac{d_{P,h}^{\pi^\star}(s_h^k, a_h^k)}{d_{P,h}^{\mu}(s_h^k, a_h^k)} \left( \sigma_{h,s_h^k,a_h^k}(V_{h+1}^\star) - \sigma_{h,s_h^k,a_h^k}(V_{h+1}^k) \right)$$

$$- \left( 1 + \frac{1}{H} \right) \sum_{s \in \mathcal{S}} d_{P,h+1}^{\pi^\star}(s) \left( V_{h+1}^\star(s) - V_{h+1}^k(s) \right), \tag{99}$$

and let us consider

$$W_{h+1}^k(s,a) := \sigma_{h,s_h^k,a_h^k}(V_{h+1}^\star) - \sigma_{h,s_h^k,a_h^k}(V_{h+1}^k), \qquad C_{\mathrm{d}} := \left( 1 + \frac{1}{H} \right) \leq 2 \tag{100}$$

which satisfies

$$\left\| W_{h+1}^k(s,a) \right\|_\infty \leq \left\| V_{h+1}^\star \right\|_\infty + \left\| V_{h+1}^k \right\|_\infty \leq 2H =: C_{\mathrm{w}}. \tag{101}$$

Similarly, in view of Lemma 11, we can show that with probability at least $1 - \delta$,

$$\left| \sum_{k=1}^{K} \left( B_{h,k} - Z_{h,k} \right) \right| = \left| \sum_{k=1}^{K} X_{h,k} \right|$$

$$\leq 16\sqrt{H^2 C^\star K \log \frac{2H}{\delta}} + 8HC^\star \log \frac{2H}{\delta}. \tag{102}$$

*Step 3: putting all this together.* Substitution results in equation 98 and equation 102 back into equation 95 completes the proof of equation 93 as follows

$$A_h \leq \sum_{k=1}^{K} B_{h,k} + \left| \sum_{k=1}^{K} \left( Z_{h,k} - B_{h,k} \right) \right| + \left| \sum_{k=1}^{K} \left( A_{h,k} - Y_{h,k} \right) \right|$$

$$\leq \sum_{k=1}^{K} B_{h,k} + 24\sqrt{H^2 C^\star K \log \frac{2H}{\delta}} + 12HC^\star \log \frac{2H}{\delta}.$$

This hence completes the proof. $\qquad \square$

**Lemma 14.** *Denote the term* $\sum_{k=1}^{K} \sum_{(s,a) \in \mathcal{S} \times \mathcal{A}} d_{P,h}^{\pi^\star}(s,a) \eta_0^{N_h^k(s,a)} H + 2\sum_{k=1}^{K} \sum_{(s,a) \in \mathcal{S} \times \mathcal{A}} d_{P,h}^{\pi^\star}(s,a) \sum_{n=1}^{N_h^k(s,a)} \eta_n^{N_h^k(s,a)} b_n$ *by* $I_h$. *Consider any* $\delta \in (0,1)$. *With probability at least* $1 - \delta$, *we have*

$$\sum_{h=1}^{H} \left( 1 + \frac{1}{H} \right)^{h-1} \left( I_h + 24\sqrt{H^2 C^\star K \log \frac{2H}{\delta}} + 12HC^\star \log \frac{2H}{\delta} \right)$$

$$\lesssim H^2 S C^\star \iota + \sqrt{H^5 S C^\star K \iota^3}, \tag{103}$$

*where we recall that* $\iota := \log \left( \frac{SAT}{\delta} \right)$.

*Proof.* The proof can be obtained by directly following the proof in (Shi et al., 2022), and is hence omitted here.

$\square$

### D.3    PROOF OF THEOREM 2

We then proceed to the proof.

**Theorem 15.** *(Restatement of Theorem 2) Consider any $\delta \in (0, 1)$. Suppose that the behavior policy $\mu$ satisfies Assumption 1. There exists some universal constant $c_a$, such that if we set $\iota := \log\left(\frac{SAT}{\delta}\right)$ and set $T > SC^\star\iota$, then the policy $\widehat{\pi}$ returned by Algorithm 1 satisfies*

$$V_1^\star(\rho) - V_1^{\widehat{\pi}}(\rho) \leq c_a\sqrt{\frac{H^6 SC^\star\iota^3}{T}} \tag{104}$$

*with probability at least $1 - \delta$.*

*Proof.* For any state-action pair $(s, a)$, according to the update rule specified in Algorithm 1, we have

$$\begin{aligned}
Q_h^k(s, a) &= Q_h^{k^{N_h^k}+1}(s, a)\\
&= \left(1 - \eta_{N_h^k}\right)Q_h^{k^{N_h^k}}(s, a) + \eta_{N_h^k}\left\{r_h(s, a) + V_{h+1}^{k^{N_h^k}}\left(s_{h+1}^{k^{N_h^k}}\right) - R_{s,a}\kappa(V_{h+1}^{k^{N_h^k}}) - b_{N_h^k}\right\},
\end{aligned} \tag{105}$$

where the first identity holds because $k^{N_h^k}$ denotes the most recent episode before $k$ that visits $(s, a)$ at step $h$, and the learning rate is defined as in equation 55. Note that $k > k^{N_h^k}$ always holds. Applying the above relation recursively and using the notation defined in equation 56, we obtain

$$Q_h^k(s, a) = \eta_0^{N_h^k}Q_h^1(s, a) + \sum_{n=1}^{N_h^k}\eta_n^{N_h^k}\left(r_h(s, a) + V_{h+1}^{k^n}\left(s_{h+1}^{k^n}\right) - R_{s,a}\kappa(V_{h+1}^{k^n}) - b_n\right). \tag{106}$$

Applying Lemma 12, the optimality gap term equation 104 can be decomposed as follows

$$\begin{aligned}
&V_1^\star(\rho) - V_1^{\widehat{\pi}}(\rho)\\
&= \mathbb{E}_{s_1\sim\rho}\left[V_1^\star(s_1)\right] - \mathbb{E}_{s_1\sim\rho}\left[V_1^{\pi^K}(s_1)\right]\\
&\overset{(a)}{\leq} \mathbb{E}_{s_1\sim\rho}\left[V_1^\star(s_1)\right] - \mathbb{E}_{s_1\sim\rho}\left[V_1^K(s_1)\right]\\
&\overset{(b)}{\leq} \frac{1}{K}\sum_{k=1}^K\left(\mathbb{E}_{s_1\sim\rho}\left[V_1^\star(s_1)\right] - \mathbb{E}_{s_1\sim\rho}\left[V_1^k(s_1)\right]\right)\\
&= \frac{1}{K}\sum_{k=1}^K\sum_{s\in\mathcal{S}}d_1^{\pi^*}(s)\left(V_1^\star(s) - V_1^k(s)\right),
\end{aligned} \tag{107}$$

where (a) follows from Lemma 12 (i.e., $V_1^{\pi^K}(s) \geq V_1^K(s)$ for all $s \in \mathcal{S}$), (b) results from the monotonicity property in Lemma 12, and the final equality holds because $d_1^{\pi^*}(s) = \rho(s)$.

We then bound the right-hand side of equation 107. Since $\pi^\star$ is a deterministic policy, $d_{P,h}^{\pi^*}(s) = d_{P,h}^{\pi^*}(s, \pi^\star(s))$. And from the fact that $V_h^k(s) \geq \max_a Q_h^k(s, a) \geq Q_h^k(s, \pi_h^\star(s))$ and $V_h^\star(s) = Q_h^\star(s, \pi_h^\star(s))$, we have that

$$\begin{aligned}
&\sum_{k=1}^K\sum_{s\in\mathcal{S}}d_{P,h}^{\pi^*}(s)\left(V_h^\star(s) - V_h^k(s)\right)\\
&= \sum_{k=1}^K\sum_{s\in\mathcal{S}}d_{P,h}^{\pi^*}(s, \pi_h^\star(s))\left(V_h^\star(s) - V_h^k(s)\right)
\end{aligned}$$

$$\leq \sum_{k=1}^{K} \sum_{s \in \mathcal{S}} d_{P,h}^{\pi^{\star}}(s, \pi_h^{\star}(s)) \left( Q_h^{\star}(s, \pi_h^{\star}(s)) - Q_h^k(s, \pi_h^{\star}(s)) \right)$$

$$= \sum_{k=1}^{K} \sum_{(s,a) \in \mathcal{S} \times \mathcal{A}} d_{P,h}^{\pi^{\star}}(s, a) \left( Q_h^{\star}(s, a) - Q_h^k(s, a) \right), \tag{108}$$

for any $h \in [H]$, where the last identity holds because

$$d_{P,h}^{\pi^{\star}}(s, a) = 0 \qquad \text{for any } a \neq \pi_h^{\star}(s). \tag{109}$$

To further bound the term $Q_h^{\star}(s, a) - Q_h^k(s, a)$ in equation 108, we first adapt equation 58 and have that

$$Q_h^{\star}(s, a) = \sum_{n=0}^{N_h^k} \eta_n^{N_h^k} Q_h^{\star}(s, a)$$

$$= \eta_0^{N_h^k} Q_h^{\star}(s, a) + \sum_{n=1}^{N_h^k} \eta_n^{N_h^k} Q_h^{\star}(s, a)$$

$$= \eta_0^{N_h^k} Q_h^{\star}(s, a) + \sum_{n=1}^{N_h^k} \eta_n^{N_h^k} \left( r_h(s, a) + \sigma_{h,s,a}(V_{h+1}^{\star}) \right), \tag{110}$$

where the second line follows from the robust Bellman's optimality equation. Combining equation 106 and equation 110 implies that

$$Q_h^{\star}(s, a) - Q_h^k(s, a)$$

$$= \eta_0^{N_h^k} \left( Q_h^{\star}(s, a) - Q_h^1(s, a) \right) + \sum_{n=1}^{N_h^k} \eta_n^{N_h^k} \left( \sigma_{h,s,a}(V_{h+1}^{\star}) - V_{h+1}^{k^n}(s_{h+1}^{k^n}) + R_{s,a}\kappa(V_{h+1}^{k^n}) + b_n \right)$$

$$\overset{(a)}{=} \eta_0^{N_h^k} \left( Q_h^{\star}(s, a) - Q_h^1(s, a) \right) + \sum_{n=1}^{N_h^k} \eta_n^{N_h^k} b_n + \sum_{n=1}^{N_h^k} \eta_n^{N_h^k} \left( \sigma_{h,s,a}(V_{h+1}^{\star}) - \sigma_{h,s,a}(V_{h+1}^{k^n}) \right)$$

$$+ \sum_{n=1}^{N_h^k} \eta_n^{N_h^k} \left( P_{h,s,a} - P_h^{k^n} \right) V_{h+1}^{k^n} \tag{111}$$

$$\leq \eta_0^{N_h^k} H + 2 \sum_{n=1}^{N_h^k} \eta_n^{N_h^k} b_n + \sum_{n=1}^{N_h^k} \eta_n^{N_h^k} \left( \sigma_{h,s,a}(V_{h+1}^{\star}) - \sigma_{h,s,a}(V_{h+1}^{k^n}) \right), \tag{112}$$

where (a) is from Lemma 6 and the definition of $P_h^{k^n} V_{h+1}^{k^n} = V_{h+1}^{k^n}(s_{h+1}^{k^n})$, and the last inequality follows from the fact $Q_h^{\star}(s, a) - Q_h^1(s, a) = Q_h^{\star}(s, a) - 0 \leq H$ and equation 71 in Lemma 12. Plug equation 112 in equation 108, we have that

$$\sum_{k=1}^{K} \sum_{s \in \mathcal{S}} d_{P,h}^{\pi^{\star}}(s) \left( V_h^{\star}(s) - V_h^k(s) \right)$$

$$\leq \underbrace{\sum_{k=1}^{K} \sum_{(s,a) \in \mathcal{S} \times \mathcal{A}} d_{P,h}^{\pi^{\star}}(s, a) \eta_0^{N_h^k(s,a)} H + 2 \sum_{k=1}^{K} \sum_{(s,a) \in \mathcal{S} \times \mathcal{A}} d_{P,h}^{\pi^{\star}}(s, a) \sum_{n=1}^{N_h^k(s,a)} \eta_n^{N_h^k(s,a)} b_n}_{=: I_h}$$

$$+ \sum_{k=1}^{K} \sum_{(s,a) \in \mathcal{S} \times \mathcal{A}} d_{P,h}^{\pi^{\star}}(s, a) \sum_{n=1}^{N_h^k(s,a)} \eta_n^{N_h^k(s,a)} \left( \sigma_{h,s,a}(V_{h+1}^{\star}) - \sigma_{h,s,a}(V_{h+1}^{k^n}) \right). \tag{113}$$

We then bound the last term on the right-hand side of equation 113. By applying Lemma 13, it implies that

$$
\sum_{k=1}^{K}\sum_{s\in\mathcal{S}} d_{P,h}^{\pi^\star}(s)\left(V_h^\star(s) - V_h^k(s)\right)
$$

$$
\leq \left(1 + \frac{1}{\Gamma}\right)\sum_{k=1}^{K}\sum_{s\in\mathcal{S}} d_{P,h+1}^{\pi^\star}(s)\left(V_{h+1}^\star(s) - V_{h+1}^k(s)\right)
$$

$$
+ I_h + 24\sqrt{H^2 C^\star K \log\frac{2H}{\delta}} + 12HC^\star \log\frac{2H}{\delta}. \tag{114}
$$

Recursively applying equation 114 over the time steps $h = H, H-1, \cdots, 1$ with the terminal condition $V_{H+1}^k = V_{H+1}^\star = 0$ further implies that

$$
\sum_{k=1}^{K}\sum_{s\in\mathcal{S}} d_1^{\pi^\star}(s)\left(V_1^\star(s) - V_1^k(s)\right)
$$

$$
\leq \max_{h\in[H]}\sum_{k=1}^{K}\sum_{s\in\mathcal{S}} d_{P,h}^{\pi^\star}(s)\left(V_h^\star(s) - V_h^k(s)\right)
$$

$$
\leq \sum_{h=1}^{H}\left(1 + \frac{1}{\Gamma}\right)^{h-1}\left(I_h + 24\sqrt{H^2 C^\star K \log\frac{2H}{\delta}} + 12HC^\star \log\frac{2H}{\delta}\right). \tag{115}
$$

Finally, to bound the right-hand side of equation 115, we combine Lemma 14 and equation 107, which yields

$$
V_1^\star(\rho) - V_1^{\widehat{\pi}}(\rho)
$$

$$
\leq \frac{1}{K}\sum_{k=1}^{K}\sum_{s\in\mathcal{S}} d_1^{\pi^\star}(s)\left(V_1^\star(s) - V_1^k(s)\right)
$$

$$
\leq \frac{1}{K}\max_{h\in[H]}\sum_{k=1}^{K}\sum_{s\in\mathcal{S}} d_{P,h}^{\pi^\star}(s)\left(V_h^\star(s) - V_h^k(s)\right)
$$

$$
\leq \frac{c_{\mathrm{a}}}{2}\sqrt{\frac{H^5 SC^\star \iota^3}{K}} + \frac{c_{\mathrm{a}}}{2}\frac{H^2 SC^\star \iota}{K} = \frac{c_{\mathrm{a}}}{2}\sqrt{\frac{H^6 SC^\star \iota^3}{T}} + \frac{c_{\mathrm{a}}}{2}\frac{H^3 SC^\star \iota}{T}
$$

$$
\leq c_{\mathrm{a}}\sqrt{\frac{H^6 SC^\star \iota^3}{T}} \tag{116}
$$

for some sufficiently large constant $c_{\mathrm{a}} > 0$, where the last inequality is valid as long as $T > SC^\star \iota$.

This hence completes the proof of Theorem 2. $\qquad\square$

# E  ANALYSIS OF THE INFINITE HORIZON SETTING

## E.1  ALGORITHM FOR INFINITE HORIZON

In this section, we present the analysis of the infinite horizon robust MDPs.

## E.2  NOTATION

The notation used in the proof for the infinite horizon setting is largely similar to that used in the finite horizon case. For any state $s \in \mathcal{S}$ and action $a \in \mathcal{A}$, we define:

$$
P_{s,a} = P(\cdot \mid s, a) \in \mathbb{R}^{1\times S}
$$

to be the $(s, a)$-th row of a probability transition matrix $P \in \mathbb{R}^{SA\times S}$.

---

**Algorithm 3** Double-Pessimism Q-Learning for infinite-horizon RMDPs.

---

**Input:** $\mathcal{D}$, target success probability $1 - \delta$, uncertainty set radius $R$, $\Gamma = \left\lceil \frac{4}{1-\gamma} \log \frac{ST}{\delta} \right\rceil$, penalty
function $\kappa$
**Initialize:** $Q_0(s,a) = 0$, $V_0(s) = 0$, $n_0(s,a) = 0$, $\forall s, a$
**for** $t = 1, ..., T$ **do**
    Sample a sample $(s_{t-1}, a_{t-1}, s_t)$ from $\mathcal{D}$
    $n_t(s_{t-1}, a_{t-1}) \leftarrow n_{t-1}(s_{t-1}, a_{t-1}) + 1$; $n_t(s,a) \leftarrow n_{t-1}(s,a), \forall (s,a) \neq (s_{t-1}, a_{t-1})$
    $n \leftarrow n_t(s,a)$; $\eta_n \leftarrow (\Gamma + 1)/(\Gamma + n)$
    $b_n \leftarrow c_b \sqrt{\frac{\Gamma \log(SAT/\delta)}{n(1-\gamma)^2}}$
    $Q_t(s_{t-1}, a_{t-1}) = (1 - \eta_n) Q_{t-1}(s_{t-1}, a_{t-1}) + \eta_n \Big\{ r(s_{t-1}, a_{t-1}) + \gamma V_{t-1}(s_t) - \gamma \kappa_{s_{t-1}, a_{t-1}}(V_{t-1}) - b_n \Big\}$
    $Q_t(s,a) = Q_{t-1}(s,a)$ for all $(s,a) \neq (s_{t-1}, a_{t-1})$
    $V_t(s_{t-1}) = \max \Big\{ \max_{a \in \mathcal{A}} Q_t(s_{t-1}, a), V_{t-1}(s_{t-1}) \Big\}$,
    $V_t(s) = V_{t-1}(s)$ for all $s \neq s_{t-1}$.
**end for**
$\widehat{\pi}(s) = \arg\max_{a \in \mathcal{A}} Q_T(s,a), \forall s$
**Output:** $\widehat{\pi}$

---

For any $t \geq 0$, we define $P_t \in \mathbb{R}^{SA \times S}$ to be an empirical probability transition matrix, given by:

$$P_t(s' \mid s, a) = \begin{cases} 1, & \text{if } (s, a, s') = (s_{t-1}, a_{t-1}, s_t) \\ 0, & \text{otherwise} \end{cases} \tag{117}$$

for all $s, s' \in \mathcal{S}$ and $a \in \mathcal{A}$.

For any deterministic policy $\pi$, we introduce two probability transition kernels: $P_\pi : \mathcal{S} \to \Delta(\mathcal{S})$ and $P^\pi : \mathcal{S} \times \mathcal{A} \to \Delta(\mathcal{S} \times \mathcal{A})$, defined as follows:

$$P_\pi(s' \mid s) = P(s' \mid s, \pi(s)), \tag{118a}$$

$$P^\pi(s', a' \mid s, a) = \begin{cases} P(s' \mid s, a), & \text{if } a' = \pi(s') \\ 0, & \text{otherwise} \end{cases} \tag{118b}$$

for any $(s, a), (s', a') \in \mathcal{S} \times \mathcal{A}$.

Additionally, we define $\rho^{\pi^\star}$ to be a distribution over $\mathcal{S} \times \mathcal{A}$ such that:

$$\rho^{\pi^\star}(s, a) = \begin{cases} \rho(s), & \text{if } a = \pi^\star(s) \\ 0, & \text{otherwise} \end{cases} \tag{119}$$

For any sequence $\{a_i\}_{i=n_1}^{n_2}$ and two integers $m_1$ and $m_2$, we define:

$$\sum_{i=m_1}^{m_2} a_i = \begin{cases} \sum_{i=\max\{n_1, m_1\}}^{\min\{n_2, m_2\}} a_i, & \text{if } \max\{n_1, m_1\} \leq \min\{n_2, m_2\} \\ 0, & \text{otherwise} \end{cases}$$

### E.3 LEMMAS FOR THEOREM 3

**Lemma 16.** *(Lemma 4.1 in (Jin et al., 2018), Lemma 1 in (Li et al., 2021)) Recall the learning rates are*

$$\eta_0^t := \prod_{j=1}^{t} (1 - \eta_j) \qquad and \qquad \eta_i^t := \begin{cases} \eta_i \prod_{j=i+1}^{t} (1 - \eta_j), & \text{if } t > i, \\ \eta_i, & \text{if } t = i, \\ 0, & \text{if } t < i, \end{cases} \tag{120}$$

*where $\eta_j = (\Gamma + 1)/(\Gamma + j)$. Then*

1. *For any integer $t \geq 1$, $\sum_{i=1}^{t} \eta_i^t = 1$ and $\eta_0^t = 0$.*

2. *For any integer $t \geq 1$ and any $1/2 \leq a \leq 1$,*

$$\frac{1}{t^a} \leq \sum_{i=1}^{t} \frac{1}{i^a} \eta_i^t \leq \frac{2}{t^a}.$$

3. *For any integer $t \geq 1$,*

$$\max_{i \in [t]} \eta_i^t \leq \frac{2\Gamma}{t} \qquad and \qquad \sum_{i=1}^{t} \left(\eta_i^t\right)^2 \leq \frac{2\Gamma}{t}.$$

4. *For any integer $i \geq 1$,*

$$\sum_{t=i}^{\infty} \eta_i^t = 1 + \frac{1}{\Gamma}.$$

We then present the following lemma to establish an upper bound on $Q^\star - Q_t$, and simultaneously justify that the value function estimate $V_t$ is always a pessimistic view of $V^{\pi_t}$ (and hence $V^\star$).

**Lemma 17.** *With probability exceeding $1 - \delta$, for all $s \in \mathcal{S}$ and $t \in [T]$, it holds that*

$$Q^\star\big(s, \pi^\star(s)\big) - Q_t\big(s, \pi^\star(s)\big) \leq \gamma \sum_{i=1}^{n} \eta_i^n \big(\sigma_{s,\pi^\star(s)}(V^\star) - \sigma_{s,\pi^\star(s)}(V_{k_i})\big) + \beta_n\big(s, \pi^\star(s)\big), \tag{121}$$

*where $n = n_t(s, \pi^\star(s))$ and we define*

$$\beta_n\big(s, \pi^\star(s)\big) \equiv \beta_n := 3c_b \sqrt{\frac{\Gamma \iota}{n \left(1 - \gamma\right)^2}};$$

*in addition, we also have*

$$V_t(s) \leq V^{\pi_t}(s) \leq V^\star(s), \qquad \forall s \in \mathcal{S}. \tag{122}$$

*Proof.* **Proof of equation 121.** Consider any given pair $(s, a) \in \mathcal{S} \times \mathcal{A}$ and denote $n = n_t(s, a)$, the total number of times that $(s, a)$ has been visited prior to time $t$. Set $k_0 = -1$, and let

$$k_i := \min \Big\{ \big\{ 0 \leq k < T : k > k_{i-1}, (s_k, a_k) = (s, a) \big\}, T \Big\} \tag{123}$$

for each $1 \leq i \leq T$. Clearly, each $k_i$ is a stopping time. In view of the update rule, we have

$$Q_t(s, a) = \sum_{i=1}^{n} \eta_i^n \Big\{ r(s, a) + \gamma V_{k_i}(s_{k_i+1}) - \gamma \kappa(V_{k_i}) - b_i(s, a) \Big\},$$

which together with the robust Bellman optimality equation gives

$$(Q^\star - Q_t)(s, a)$$

$$= r(s, a) + \gamma \sigma_{s,a}(V^\star) - \sum_{i=1}^{n} \eta_i^n \Big\{ r(s, a) + \gamma V_{k_i}(s_{k_i+1}) - \gamma \kappa(V_{k_i}) - b_i(s, a) \Big\}$$

$$= \gamma \sigma_{s,a}(V^{\star}) - \sum_{i=1}^{n} \eta_{i}^{n} \left\{ \gamma V_{k_i}(s_{k_i+1}) - \gamma \kappa(V_{k_i}) - b_i(s,a) \right\}$$

$$= \sum_{i=1}^{n} \eta_{i}^{n} \gamma \left( \sigma_{s,a}(V^{\star}) - \sigma_{s,a}(V_{k_i}) \right) + \sum_{i=1}^{n} \eta_{i}^{n} \gamma \left( (P - P_{k_i}) V_{k_i} \right)(s,a) + \sum_{i=1}^{n} \eta_{i}^{n} b_i(s,a), \quad (124)$$

where the last two lines are valid since $\sum_{i=1}^{n} \eta_{i}^{n} = 1$ (cf. Lemma 16) and Lemma 6.

Henceforth, we only focus on the case where $a = \pi^{\star}(s)$. Define $\mathcal{F}_i$ to be the $\sigma$-field generated by $\{(s_i, a_i)\}_{i=0}^{k_i}$. It is straightforward to check that for any $1 \leq \tau \leq T$,

$$\left\{ \mathbf{1}_{k_i < T} \left( (P - P_{k_i}) V_{k_i} \right)(s, \pi^{\star}(s)) \right\}_{i=1}^{\tau}$$

is a martingale difference sequence with respect to $\{\mathcal{F}_i\}_{i \geq 0}$. Then, we can invoke the Azuma-Hoeffding inequality together with the basic bound $\|V_{k_i}\|_{\infty} \leq \frac{1}{1-\gamma}$ to show that for any fixed $s \in \mathcal{S}$ and $\tau \in [T]$,

$$\left| \sum_{i=1}^{\tau} \mathbf{1}_{k_i < T} \eta_{i}^{\tau} \left( (P - P_{k_i}) V_{k_i} \right)(s, \pi^{\star}(s)) \right| \lesssim \frac{1}{1-\gamma} \sqrt{\sum_{i=1}^{\tau} (\eta_{i}^{\tau})^2 \log \frac{ST}{\delta}}$$

$$\lesssim \sqrt{\frac{\Gamma}{\tau (1-\gamma)^2} \log \frac{ST}{\delta}}$$

holds with probability exceeding $1 - \delta/(ST)$. Here, the last line utilizes Lemma 16. Taking the union bound over $\tau \leq T$ allows us to replace $\tau$ with $n = n_t(s,a)$ in the above inequality, namely, for any fixed $s \in \mathcal{S}$ and $a \in \mathcal{A}$, with probability exceeding $1 - \delta/S$ we have

$$\left| \sum_{i=1}^{n} \eta_{i}^{n} \gamma \left( (P - P_{k_i}) V_{k_i} \right)(s, \pi^{\star}(s)) \right| \lesssim \sqrt{\frac{\Gamma \iota}{n (1-\gamma)^2}} \quad (125)$$

holds for all $n = n_t(s, \pi^{\star}(s))$ with $1 \leq t \leq T$. In view of Lemma 16, for any $s \in \mathcal{S}$ and $a \in \mathcal{A}$ we know that

$$c_b \sqrt{\frac{\Gamma \iota}{n_t(s,a)(1-\gamma)^2}} \leq \sum_{i=1}^{n_t(s,a)} \eta_{i}^{n_t(s,a)} b_i(s,a) \leq 2 c_b \sqrt{\frac{\Gamma \iota}{n_t(s,a)(1-\gamma)^2}}. \quad (126)$$

Therefore, when $c_b$ is sufficiently large, it follows that

$$(Q^{\star} - Q_t)(s, \pi^{\star}(s)) \leq \gamma \sum_{i=1}^{n} \eta_{i}^{n} \left( \sigma_{s,\pi^{\star}(s)}(V^{\star}) - \sigma_{s,\pi^{\star}(s)}(V_{k_i}) \right) + 3 c_b \sqrt{\frac{\Gamma \iota}{n (1-\gamma)^2}}.$$

Taking the union bound over $s \in \mathcal{S}$ and defining

$$\beta_n(s, \pi^{\star}(s)) := 3 c_b \sqrt{\frac{\Gamma \iota}{n (1-\gamma)^2}},$$

we can conclude that with probability exceeding $1 - \delta$,

$$(Q^{\star} - Q_t)(s, \pi^{\star}(s)) \leq \gamma \sum_{i=1}^{n} \eta_{i}^{n} \left( \sigma_{s,\pi^{\star}(s)}(V^{\star}) - \sigma_{s,\pi^{\star}(s)}(V_{k_i}) \right) + \beta_n(s, \pi^{\star}(s))$$

for all $s \in \mathcal{S}$ and $t \in [T]$.

**Proof of equation 122.** Note that $V^{\star} \geq V^{\pi_t}$ holds trivially due to the optimality of $V^{\star}$. We are therefore left with showing $V^{\pi_t} \geq V_t$. Suppose for the moment that with probability exceeding $1 - \delta$, for all $s \in \mathcal{S}$, $t \in [T]$ and $j \in [t]$, it holds that

$$(Q^{\pi_t} - Q_j)(s, \pi_t(s)) \geq \gamma \left( \sigma_{s,\pi_t(s)}(V^{\pi_t}) - \sigma_{s,\pi_t(s)}(V_j) \right) \mathbf{1}\left\{ n_t(s, \pi_t(s)) \geq 1 \right\}; \quad (127)$$

the proof of this claim (127) is deferred to later. As a consequence, for every $s \in \mathcal{S}$ and $t \in [T]$, there exists $j(t) \in [t]$ such that

$$
\begin{aligned}
\left(V^{\pi_t} - V_t\right)(s) & \overset{(a)}{=} Q^{\pi_t}\left(s, \pi_t(s)\right) - Q_{j(t)}\left(s, \pi_t(s)\right) \\
& \overset{(b)}{=} Q^{\pi_t}\left(s, \pi_t(s)\right) - Q_{j(t)}\left(s, \pi_{j(t)}(s)\right) \\
& \overset{(c)}{\geq} \min\left\{\gamma\left(\sigma_{s,\pi_t(s)}(V^{\pi_t}) - \sigma_{s,\pi_t(s)}(V_{j(t)})\right), 0\right\} \\
& \overset{(d)}{\geq} \min\left\{\gamma\left(\sigma_{s,\pi_{j(t)}(s)}(V^{\pi_t}) - \sigma_{s,\pi_{j(t)}(s)}(V_t)\right), 0\right\}.
\end{aligned}
$$

Here, (a) and (b) hold since the update rule asserts that there must exist some $j(t) \leq t$ such that $V_t(s) = V_{j(t)}(s) = Q_{j(t)}(s, \pi_{j(t)}(s))$ and $\pi_t(s) = \pi_{j(t)}(s)$; (c) utilizes (127); and (d) follows from the monotonicity of $V_t$ in $t$ (by construction). By setting

$$
s_{\min} := \arg\min_{s \in \mathcal{S}} \left(V^{\pi_t} - V_t\right)(s),
$$

we can deduce that

$$
\begin{aligned}
\left(V^{\pi_t} - V_t\right)(s_{\min}) & \geq \min\left\{\gamma\left(\sigma_{s_{\min}, \pi_{j(t)}(s_{\min})}(V^{\pi_t}) - \sigma_{s_{\min}, \pi_{j(t)}(s_{\min})}(V_t)\right), 0\right\} \\
& \geq \min\left\{\gamma \min_{s \in \mathcal{S}}\left(V^{\pi_t} - V_t\right)(s), 0\right\} \\
& = \min\left\{\gamma\left(V^{\pi_t} - V_t\right)(s_{\min}), 0\right\},
\end{aligned}
$$

which together with the assumption $0 < \gamma < 1$ immediately gives

$$
\left(V^{\pi_t} - V_t\right)(s_{\min}) \geq 0.
$$

Given that $\left(V^{\pi_t} - V_t\right)(s) \geq \left(V^{\pi_t} - V_t\right)(s_{\min})$ for every $s \in \mathcal{S}$, we conclude the proof.

**Now we show equation 127.** First of all, if $n_t\left(s, \pi_t(s)\right) = 0$, then for all $j \in [t]$, $Q_j\left(s, \pi_t(s)\right) = 0$ since it is never updated; therefore, (127) holds true. From now on, we shall only focus on the case when $n_t\left(s, \pi_t(s)\right) \geq 1$.

Consider any $s \in \mathcal{S}$, $t \in [T]$ and $j \in [t]$. For the moment, let us define $\{k_i\}_{i=1}^{T}$ w.r.t. the state-action pair $\left(s, \pi_t(s)\right)$ in the same way as (123). We can then repeat the argument in (124) to decompose

$$
\left(Q^{\pi_t} - Q_j\right)\left(s, \pi_t(s)\right)
$$

$$
= \left(r + \gamma\sigma(V^{\pi_t})\right)\left(s, \pi_t(s)\right) - \sum_{i=1}^{n_j(s,\pi_t(s))} \eta_i^{n_j(s,\pi_t(s))}\left\{r\left(s, \pi_t(s)\right) + \gamma V_{k_i}\left(s_{k_i+1}\right) - R_s^{\pi_t(s)}\kappa(V^{\pi_t}) - b_i\left(s, \pi_t(s)\right)\right\}
$$

$$
= \sum_{i=1}^{n_j(s,\pi_t(s))} \eta_i^{n_j(s,\pi_t(s))}\gamma\left\{\left(\sigma_{s,\pi_t(s)}(V^{\pi_t}) - \sigma_{s,\pi_t(s)}(V_{k_i})\right) + \left(\left(P - P_{k_i}\right)V_{k_i}\right)\left(s, \pi_t(s)\right)\right\}
$$

$$
+ \sum_{i=1}^{n_j(s,\pi_t(s))} \eta_i^{n_j(s,\pi_t(s))}b_i\left(s, \pi_t(s)\right)
$$

$$
\geq \left(\sum_{i=1}^{n_j(s,\pi_t(s))} \eta_i^{n_j(s,\pi_t(s))}\right)\gamma \min_{1 \leq i \leq n}\left(\sigma_{s,\pi_t(s)}(V^{\pi_t}) - \sigma_{s,\pi_t(s)}(V_{k_i})\right)
$$

$$
+ \sum_{i=1}^{n_j(s,\pi_t(s))} \eta_i^{n_j(s,\pi_t(s))}\gamma\left(\left(P - P_{k_i}\right)V_{k_i}\right)\left(s, \pi_t(s)\right)
$$

$$
+ \sum_{i=1}^{n_j(s,\pi_t(s))} \eta_i^{n_j(s,\pi_t(s))}b_i\left(s, \pi_t(s)\right)
$$

$$
\geq \gamma\left(\sigma_{s,\pi_t(s)}(V^{\pi_t}) - \sigma_{s,\pi_t(s)}(V_t)\right)
$$

$$+ \sum_{i=1}^{n_j(s,\pi_t(s))} \eta_i^{n_j(s,\pi_t(s))} \gamma \Big( \big( P - P_{k_i} \big) V_{k_i} \Big) \big( s, \pi_t(s) \big) + c_b \sqrt{\frac{\Gamma \iota}{n_j \big( s, \pi_t(s) \big) \left( 1 - \gamma \right)^2}}.$$

Here, the last inequality follows from (126), as well as the facts that $\sum_{i=1}^{n_j(s,\pi_t(s))} \eta_i^{n_j(s,\pi_t(s))} = 1$ (cf. Lemma 16) and that $V_t$ is non-decreasing in $t$. It thus boils down to showing that for every $s \in \mathcal{S}$, $t \in [T]$ and $j \in [t]$,

$$\sum_{i=1}^{n_j(s,\pi_t(s))} \eta_i^{n_j(s,\pi_t(s))} \gamma \Big( \big( P - P_{k_i} \big) V_{k_i} \Big) \big( s, \pi_t(s) \big) \lesssim \sqrt{\frac{\Gamma \iota}{n_j \big( s, \pi_t(s) \big) \left( 1 - \gamma \right)^2}}. \tag{128}$$

If this were true and if $c_b$ is sufficiently large, then we could combine the above two inequalities to conclude the proof of (127).

We then prove the inequality equation 128. Notice that for all $(s, \pi_t(s))$ such that $n_t(s, \pi_t(s)) \geq 1$, it must appear at least once in the sample trajectory. Therefore it suffices to show that for all $0 \leq l < T$ and $t \in [T]$, it holds that

$$\sum_{i=1}^{n_t(s_l,a_l)} \eta_i^{n_t(s_l,a_l)} \gamma \Big( \big( P - P_{k_i} \big) V_{k_i} \Big) \big( s_l, a_l \big) \lesssim \sqrt{\frac{\Gamma \iota}{n_t(s_l, a_l) \left( 1 - \gamma \right)^2}},$$

where we abuse the notation by defining $\{k_i\}_{i=1}^T$ for the state-action pair $(s_l, a_l)$ in the same way as (123). Furthermore, it suffices to only check those $(s_l, a_l)$ in the sample trajectory that were visited for the first time, i.e., $n_l(s_l, a_l) = 0$ and $n_{l+1}(s_l, a_l) = 1$. It is straightforward to check that, for any $1 \leq \tau \leq T$,

$$\left\{ \mathbf{1}_{k_i < T} \Big( \big( P - P_{k_i} \big) V_{k_i} \Big) \big( s_l, a_l \big) \right\}_{i=1}^{\tau}$$

is a martingale difference sequence with respect to $\{\mathcal{F}_i\}_{i \geq 0}$, where $\mathcal{F}_i$ is the $\sigma$-field generated by $\{(s_i, a_i)\}_{i=0}^{k_i}$. Then, we can invoke the Azuma-Hoeffding inequality to show that: for any such $(s_l, a_l)$ and any $\tau \in [T]$, with probability exceeding $1 - \delta/T^2$,

$$\left| \sum_{i=1}^{\tau} \mathbf{1}_{k_i < T} \eta_i^{\tau} \Big( \big( P - P_{k_i} \big) V_{k_i} \Big) \big( s_l, a_l \big) \right| \lesssim \frac{1}{1 - \gamma} \sqrt{\sum_{i=1}^{\tau} (\eta_i^{\tau})^2 \log \frac{T}{\delta}} \lesssim \sqrt{\frac{\Gamma \iota}{\tau \left( 1 - \gamma \right)^2}}.$$

Taking the union bound over $\tau \in [T]$ allows us to replace $\tau$ with $n_t(s_l, a_l)$ in the above inequality, namely, this shows that for any such $(s_l, a_l)$, with probability exceeding $1 - \delta/T$ we have

$$\left| \sum_{i=1}^{n_t(s_l,a_l)} \eta_i^{n_t(s_l,a_l)} \Big( \big( P - P_{k_i} \big) V_{k_i} \Big) \big( s_l, a_l \big) \right| \lesssim \sqrt{\frac{\Gamma \iota}{n_t(s_l, a_l) \left( 1 - \gamma \right)^2}}$$

for all $t \in [T]$. Taking the union bound over all such $(s_l, a_l)$ (which are concerned with at most $T$ pairs), we see that with probability exceeding $1 - \delta$,

$$\left| \sum_{i=1}^{n_t(s_l,a_l)} \eta_i^{n_t(s_l,a_l)} \Big( \big( P - P_{k_i} \big) V_{k_i} \Big) \big( s_l, a_l \big) \right| \lesssim \sqrt{\frac{\Gamma \iota}{n_t(s_l, a_l) \left( 1 - \gamma \right)^2}}$$

is valid for any $0 \leq j < T$ and any $t \in [T]$. This establishes the inequality equation 128, thus concluding the proof. $\qquad \square$

Next, we define two disjoint sets of state-action pairs, divided based on the associated occupancy probability induced by the behavior policy:

$$\mathcal{I} := \left\{ \big( s, \pi^\star(s) \big) \mid s \in \mathcal{S}, \mu_{\mathsf{b}} \big( s, \pi^\star(s) \big) \geq \frac{\delta}{ST} \right\}, \tag{129a}$$

$$\mathcal{I}^c := \left\{ \big( s, \pi^\star(s) \big) \mid s \in \mathcal{S}, \mu_{\mathsf{b}} \big( s, \pi^\star(s) \big) < \frac{\delta}{ST} \right\}. \tag{129b}$$

It turns out that the state-action pairs in $\mathcal{I}^c$ are rarely visited, as formalized by the following lemma.

**Lemma 18.** *(Lemma 3 in (Yan et al., 2022))With probability exceeding $1 - \delta$, we have*

$$\mathcal{I}^c \cap \left\{ (s_t, a_t) \right\}_{t=t_{\mathsf{mix}}(\delta)}^{T} = \varnothing.$$

**Lemma 19.** *(Lemma 5 in (Yan et al., 2022)) We can construct an auxiliary set of random variables* $\left\{ \left( s_k^i, a_k^i \right) : 1 \leq k \leq K - 1 \right\}$ *satisfying*

$$\left\{ \left( s_k^i, a_k^i \right) : 1 \leq k \leq K - 1 \right\} \overset{\mathsf{i.i.d.}}{\sim} \mu_{\mathsf{b}}, \tag{130a}$$

$$\mathbb{P}\left\{ \left( s_k^i, a_k^i \right) = (s_{k\tau+i}, a_{k\tau+i}) \quad \text{for all } 1 \leq k \leq K - 1 \right\} \geq 1 - \frac{\delta}{T}, \tag{130b}$$

*and*

$$\left( s_k^i, a_k^i \right) \text{ is independent of } \left\{ (s_t, a_t) : 0 \leq t \leq (k - 1)\tau + i \right\}. \tag{130c}$$

**Lemma 20.** *(Lemma 4 in (Yan et al., 2022)) Let* $\Gamma = \left\lceil \frac{4}{1-\gamma} \log \frac{ST}{\delta} \right\rceil$ *for some* $0 < \delta < 1$. *For any vector with non-negative entries* $V \in \mathbb{R}^d$, *we have*

$$\sum_{j=0}^{\infty} \left[ \gamma \left( 1 + \frac{1}{\Gamma} \right)^3 \right]^j \left\langle \rho(P_{\pi^\star})^j, V \right\rangle \lesssim \frac{1}{1-\gamma} \left\langle d_\rho^\star, V \right\rangle + \frac{\delta}{ST^4 (1-\gamma)} \|V\|_\infty. \tag{131}$$

### E.4 PROOF OF THEOREM 3

Following (Yan et al., 2022), we similarly define the following terms first:

$$\alpha_j := \left[ \gamma \left( 1 + \frac{1}{\Gamma} \right)^3 \right]^j \sum_{t=1}^{T} \left\langle \rho(P_{\pi^\star})^j, V^\star - V_t \right\rangle,$$

$$\theta_j := \left[ \gamma \left( 1 + \frac{1}{\Gamma} \right)^3 \right]^j \sum_{t=1}^{T} \sum_{s \in \mathcal{S}} \left[ \rho(P_{\pi^\star})^j \right] (s, \pi^\star(s)) \min \left\{ \beta_{n_t(s, \pi^\star(s))} (s, \pi^\star(s)), \frac{1}{1-\gamma} \right\},$$

$$\xi_j := \left[ \gamma \left( 1 + \frac{1}{\Gamma} \right)^3 \right]^j \sum_{t=1}^{t_{\mathsf{mix}}(\delta)} \left\langle \rho(P_{\pi^\star})^j, V^\star - V_t \right\rangle + \left[ \gamma \left( 1 + \frac{1}{\Gamma} \right)^3 \right]^{j+1} \left\langle \rho(P_{\pi^\star})^{j+1}, V^\star - V_0 \right\rangle,$$

$$\psi_j := \left[ \gamma \left( 1 + \frac{1}{\Gamma} \right)^3 \right]^j \sum_{t=t_{\mathsf{mix}}(\delta)}^{T} \left[ \sum_{s \in \mathcal{S}, a \in \mathcal{A}} \left[ \rho^{\pi^\star} (P^{\pi^\star})^j \right] (s, a) \sum_{i=1}^{n_t(s,a)} \eta_i^{n_t(s,a)} P_{s,a} \left( V^\star - V_{k_i(s,a)} \right) \right.$$

$$\left. - \left( 1 + \frac{1}{\Gamma} \right) \frac{\left[ \rho^{\pi^\star} (P^{\pi^\star})^j \right] (s_t, a_t)}{\mu_{\mathsf{b}}(s_t, a_t)} \sum_{i=1}^{n_t(s_t,a_t)} \eta_i^{n_t(s_t,a_t)} P_{s_t,a_t} \left( V^\star - V_{k_i(s_t,a_t)} \right) \right],$$

$$\phi_j := \gamma^{j+1} \left( 1 + \frac{1}{\Gamma} \right)^{3j+2} \sum_{t=0}^{T} \mathbf{1}_{(s_t,a_t) \in \mathcal{I}} \left[ \frac{\left[ \rho^{\pi^\star} (P^{\pi^\star})^j \right] (s_t, a_t)}{\mu_{\mathsf{b}}(s_t, a_t)} P_{s_t,a_t} \left( V^\star - V_t \right) \right.$$

$$\left. - \left( 1 + \frac{1}{\Gamma} \right) \sum_{s \in \mathcal{S}, a \in \mathcal{A}} \left[ \rho^{\pi^\star} (P^{\pi^\star})^j \right] (s, a) P_{s,a} \left( V^\star - V_t \right) \right],$$

where we recall the definition of $\mathcal{I}$ in equation 129.

We then proceed to the proof.

**Theorem 21.** *(Restatement of Theorem 3) Consider any* $\delta \in (0, 1)$. *Suppose that the behavior policy* $\mu$ *satisfies Assumption 2. The policy* $\widehat{\pi}$ *returned by Algorithm 3 satisfies*

$$V^\star(\rho) - V^{\widehat{\pi}}(\rho) \leq \tilde{\mathcal{O}} \left( \sqrt{\frac{C^\star S}{T(1-\gamma)^5}} + \frac{C^\star S}{T(1-\gamma)^2} + \frac{C^\star}{T(1-\gamma)^3} \right). \tag{132}$$

*with probability at least* $1 - \delta$.

*Proof.* Note that

$$V^\star(\rho) - V^{\widehat{\pi}}(\rho) = \langle \rho, V^\star - V^{\widehat{\pi}} \rangle \overset{(a)}{\leq} \langle \rho, V^\star - V_T \rangle \overset{(b)}{\leq} \frac{1}{T} \sum_{t=1}^{T} \langle \rho, V^\star - V_t \rangle \overset{(c)}{=} \frac{1}{T} \alpha_0. \tag{133}$$

Here, (a) holds true according to Lemma 17; (b) follows from the monotonicity of $V_t$ in $t$ (by construction); and (c) follows simply from the definition of $\alpha_0$. We then turn attention to bounding $\alpha_0$, towards which we observe that

$$\alpha_0 = \sum_{t=1}^{t_{\mathsf{mix}}(\delta)-1} \langle \rho, V^\star - V_t \rangle + \sum_{t=t_{\mathsf{mix}}(\delta)}^{T} \sum_{s \in \mathcal{S}} \rho(s) \min \left\{ Q^\star(s, \pi^\star(s)) - V_t(s), \frac{1}{1-\gamma} \right\}$$

$$\leq \sum_{t=1}^{t_{\mathsf{mix}}(\delta)-1} \langle \rho, V^\star - V_t \rangle + \sum_{t=t_{\mathsf{mix}}(\delta)}^{T} \sum_{s \in \mathcal{S}} \rho(s) \min \left\{ Q^\star(s, \pi^\star(s)) - Q_t(s, \pi^\star(s)), \frac{1}{1-\gamma} \right\}$$

$$\leq \sum_{t=1}^{t_{\mathsf{mix}}(\delta)} \langle \rho, V^\star - V_t \rangle + \gamma \underbrace{\sum_{t=t_{\mathsf{mix}}(\delta)}^{T} \sum_{s \in \mathcal{S}} \rho(s) \sum_{i=1}^{n_t(s, \pi^\star(s))} \eta_i^{n_t(s, \pi^\star(s))} \left( \sigma_{s, \pi^\star(s)}(V^\star) - \sigma_{s, \pi^\star(s)}(V_{k_i}) \right)}_{=:\zeta}$$

$$+ \underbrace{\sum_{t=1}^{T} \sum_{s \in \mathcal{S}} \rho(s) \min \left\{ \beta_{n_t(s, \pi^\star(s))}(s, \pi^\star(s)), \frac{1}{1-\gamma} \right\}}_{=: \frac{\theta_0}{1+R}}.$$

Here, the first identity holds since $V^\star(s) = Q^\star(s, \pi^\star(s))$ and $0 \leq V^\star(s) - V_t(s) \leq 1/(1-\gamma)$ for all $s \in \mathcal{S}$, the second line relies on the fact that $V_t(s) \geq \max_a Q_t(s, a) \geq Q_t(s, \pi^\star(s))$, while the last line invokes Lemma 17. With probability exceeding $1 - \delta$, the first term $\zeta$ can be upper bounded by

$$\zeta \leq \gamma \sum_{t=t_{\mathsf{mix}}(\delta)}^{T} \sum_{s \in \mathcal{S}} \rho(s) \sum_{i=1}^{n_t(s, \pi^\star(s))} \eta_i^{n_t(s, \pi^\star(s))} \left( \sigma_{s, \pi^\star(s)}(V^\star) - \sigma_{s, \pi^\star(s)}(V_{k_i}) \right)$$

$$= \gamma \sum_{t=t_{\mathsf{mix}}(\delta)}^{T} \sum_{s \in \mathcal{S}, a \in \mathcal{A}} \mu_{\mathsf{b}}(s, a) \frac{\rho^{\pi^\star}(s, a)}{\mu_{\mathsf{b}}(s, a)} \sum_{i=1}^{n_t(s,a)} \eta_i^{n_t(s,a)} P_{s, \pi^\star(s)}(V^\star - V_{k_i})$$

$$- \gamma \sum_{t=t_{\mathsf{mix}}(\delta)}^{T} \sum_{s \in \mathcal{S}, a \in \mathcal{A}} \mu_{\mathsf{b}}(s, a) \frac{\rho^{\pi^\star}(s, a)}{\mu_{\mathsf{b}}(s, a)} R_{s,a} \sum_{i=1}^{n_t(s,a)} \eta_i^{n_t(s,a)} \left( \kappa(V^\star) - \kappa(V_{k_i}) \right)$$

$$\leq \gamma \sum_{t=t_{\mathsf{mix}}(\delta)}^{T} \sum_{s \in \mathcal{S}, a \in \mathcal{A}} \mu_{\mathsf{b}}(s, a) \frac{\rho^{\pi^\star}(s, a)}{\mu_{\mathsf{b}}(s, a)} \sum_{i=1}^{n_t(s,a)} \eta_i^{n_t(s,a)} P_{s, \pi^\star(s)}(V^\star - V_{k_i})$$

$$+ 2\gamma \sum_{t=t_{\mathsf{mix}}(\delta)}^{T} \sum_{s \in \mathcal{S}, a \in \mathcal{A}} \mu_{\mathsf{b}}(s, a) \frac{\rho^{\pi^\star}(s, a)}{\mu_{\mathsf{b}}(s, a)} R_{s,a} \sum_{i=1}^{n_t(s,a)} \eta_i^{n_t(s,a)} (V^\star - V_{k_i}),$$

where we utilize the fact that $V^* \geq V_{k_i}$ and $\kappa$ is 1-Lipschitz. Hence we further have that $\alpha_0$

$$\leq (1+R)\gamma \sum_{t=t_{\mathsf{mix}}(\delta)}^{T} \sum_{s \in \mathcal{S}, a \in \mathcal{A}} \mu_{\mathsf{b}}(s, a) \frac{\rho^{\pi^\star}(s, a)}{\mu_{\mathsf{b}}(s, a)} \sum_{i=1}^{n_t(s,a)} \eta_i^{n_t(s,a)} P_{s, \pi^\star(s)}(V^\star - V_{k_i})$$

$$+ (1+R) \sum_{t=1}^{t_{\mathsf{mix}}(\delta)} \langle \rho, V^\star - V_t \rangle + \theta_0$$

$$\overset{(a)}{\leq} \gamma(1+R) \left( 1 + \frac{1}{\Gamma} \right) \sum_{t=t_{\mathsf{mix}}(\delta)}^{T} \mathbf{1}\{(s_t, a_t) \in \mathcal{I}\} \frac{\rho^{\pi^\star}(s_t, a_t)}{\mu_{\mathsf{b}}(s_t, a_t)} \sum_{i=1}^{n_t(s_t, a_t)} \eta_i^{n_t(s_t, a_t)} P_{s_t, a_t} \left( V^\star - V_{k_i(s_t, a_t)} \right) + \psi_0$$

$$+ (1+R) \sum_{t=1}^{t_{\mathsf{mix}}(\delta)} \langle \rho, V^\star - V_t \rangle + \theta_0$$

$$\overset{(b)}{\succeq} \gamma \left(1 + \frac{1}{\Gamma}\right) \sum_{t=t_{\mathsf{mix}}(\delta)}^{T} \mathbf{1}\{(s_t, a_t) \in \mathcal{I}\} \frac{\rho^{\pi^\star}(s_t, a_t)}{\mu_{\mathsf{b}}(s_t, a_t)} \left( \sum_{j=n_t(s_t,a_t)}^{n_T(s_t,a_t)} \eta^j_{n_t(s_t,a_t)} \right) P_{s_t, a_t} (V^\star - V_t) + \psi_0$$

$$+ (1+R) \sum_{t=1}^{t_{\mathsf{mix}}(\delta)} \langle \rho, V^\star - V_t \rangle + \theta_0$$

$$\overset{(c)}{\leq} \gamma \left(1 + \frac{1}{\Gamma}\right)^2 \sum_{t=0}^{T} \mathbf{1}\{(s_t, a_t) \in \mathcal{I}\} \frac{\rho^{\pi^\star}(s_t, a_t)}{\mu_{\mathsf{b}}(s_t, a_t)} P_{s_t, a_t} (V^\star - V_t) + \psi_0$$

$$+ (1+R) \sum_{t=1}^{t_{\mathsf{mix}}(\delta)} \langle \rho, V^\star - V_t \rangle + \theta_0$$

$$= \gamma \left(1 + \frac{1}{\Gamma}\right)^3 \sum_{t=0}^{T} \sum_{s \in \mathcal{S}, a \in \mathcal{A}} \rho^{\pi^\star}(s, a) P_{s,a} (V^\star - V_t) + \psi_0 + \phi_0 + (1+R) \sum_{t=1}^{t_{\mathsf{mix}}(\delta)} \langle \rho, V^\star - V_t \rangle + \theta_0$$

$$= \gamma \left(1 + \frac{1}{\Gamma}\right)^3 \sum_{t=0}^{T} \langle \rho P_{\pi^\star}, V^\star - V_t \rangle + \psi_0 + \phi_0 + (1+R) \sum_{t=1}^{t_{\mathsf{mix}}(\delta)} \langle \rho, V^\star - V_t \rangle + \theta_0$$

$$\leq \alpha_1 + \psi_0 + \phi_0 + \gamma \left(1 + \frac{1}{\Gamma}\right)^3 \langle \rho P_{\pi^\star}, V^\star - V_0 \rangle + (1+R) \sum_{t=1}^{t_{\mathsf{mix}}(\delta)} \langle \rho, V^\star - V_t \rangle + \theta_0,$$

where we remind the reader of our notation $\rho^{\pi^\star}$ in equation 119. Here, (a) is valid (i.e., $\rho(s_t, a_t)/\mu_{\mathsf{b}}(s, a)$ is well defined for $t \geq t_{\mathsf{mix}}(\delta)$) due to Lemma 18; (b) holds by grouping the terms in the previous line; and (c) utilizes Lemma 16 and the property that $V^\star \geq V_t$ (cf. Lemma 17). Therefore, we arrive at

$$\alpha_0 \leq \sum_{t=1}^{t_{\mathsf{mix}}(\delta)} \langle \rho, V^\star - V_t \rangle + \zeta + \theta_0$$

$$\leq (1+R) \sum_{t=1}^{t_{\mathsf{mix}}(\delta)} \langle \rho, V^\star - V_t \rangle + \alpha_1 + \psi_0 + \phi_0 + \gamma \left(1 + \frac{1}{\Gamma}\right)^3 \langle \rho P_{\pi^\star}, V^\star - V_0 \rangle + \theta_0$$

$$= \alpha_1 + \xi_0 + \theta_0 + \psi_0 + \phi_0,$$

where we have used the definition of $\xi_0$. Repeat the same argument to reach

$$\alpha_j \leq \alpha_{j+1} + \xi_j + \theta_j + \psi_j + \phi_j$$

for all $j \geq 1$. This in turn allows us to conclude that

$$\alpha_0 \leq \underbrace{\limsup_{j \to \infty} \alpha_j}_{=: \alpha} + \underbrace{\sum_{j=0}^{\infty} \xi_j}_{=: \xi} + \underbrace{\sum_{j=0}^{\infty} \theta_j}_{=: \theta} + \underbrace{\sum_{j=0}^{\infty} \psi_j}_{=: \psi} + \underbrace{\sum_{j=0}^{\infty} \phi_j}_{=: \phi}. \tag{134}$$

We will then bound the terms $\alpha$, $\xi$, $\theta$, $\psi$ and $\phi$ separately in the subsequent steps. Our proofs are similar to the ones in (Yan et al., 2022), hence we omit the repeated part.

**Bounding $\alpha$.** The bound is similar to (Yan et al., 2022). It is first observed that

$$\alpha = \limsup_{j \to \infty} \left[ \gamma \left(1 + \frac{1}{\Gamma}\right)^3 \right]^j \sum_{t=1}^{T} \langle \rho (P_{\pi^\star})^j, V^\star - V_t \rangle \leq \frac{T}{1 - \gamma} \limsup_{k \to \infty} \left[ \gamma \left(1 + \frac{1}{\Gamma}\right)^3 \right]^k = 0.$$

**Bounding $\xi$.**

By utilizing (131), it holds that

$$\xi = \sum_{t=1}^{t_{\mathsf{mix}}(\delta)} \left\{ \sum_{j=0}^{\infty} \left[ \gamma \left( 1 + \frac{1}{\Gamma} \right)^3 \right]^j \left\langle \rho P_{\pi^\star}^j, V^\star - V_t \right\rangle \right\} + \sum_{j=0}^{\infty} \left[ \gamma \left( 1 + \frac{1}{\Gamma} \right)^3 \right]^{j+1} \left\langle \rho (P_{\pi^\star})^{j+1}, V^\star - V_0 \right\rangle$$

$$\lesssim \frac{1}{1-\gamma} \sum_{t=0}^{t_{\mathsf{mix}}(\delta)} \left\langle d_\rho^\star, V^\star - V_t \right\rangle + \frac{1}{ST^4 (1-\gamma)} \frac{t_{\mathsf{mix}}(\delta) + 1}{1-\gamma}$$

$$\lesssim \frac{t_{\mathsf{mix}}}{(1-\gamma)^2} \log \frac{1}{\delta} + \frac{t_{\mathsf{mix}}}{T^4 (1-\gamma)^2} \log \frac{1}{\delta}.$$

**Bounding $\theta$.** Following (Yan et al., 2022), we have that Note that

$$\theta = \sum_{j=0}^{\infty} \left[ \gamma \left( 1 + \frac{1}{\Gamma} \right)^3 \right]^j \sum_{t=1}^{T} \sum_{s \in \mathcal{S}} \left[ \rho (P_{\pi^\star})^j \right] (s) \min \left\{ \beta_{n_t(s, \pi^\star(s))}, \frac{1}{1-\gamma} \right\}$$

$$\lesssim \frac{C^\star S t_{\mathsf{mix}} \iota}{(1-\gamma)^2} + \sqrt{\frac{C^\star S T \iota^2}{(1-\gamma)^5}}.$$

**Bounding $\psi$.** Note that

$$\psi = \sum_{j=0}^{\infty} \gamma \left[ \gamma \left( 1 + \frac{1}{\Gamma} \right)^3 \right]^j \sum_{t=t_{\mathsf{mix}}(\delta)}^{T} \left[ \sum_{s \in \mathcal{S}, a \in \mathcal{A}} \left[ \rho^{\pi^\star} (P^{\pi^\star})^j \right] (s, a) \sum_{i=1}^{n_t(s,a)} \eta_i^{n_t(s,a)} P_{s,a} \left( V^\star - V_{k_i(s,a)} \right) \right.$$

$$\left. - \left( 1 + \frac{1}{\Gamma} \right) \frac{\left[ \rho^{\pi^\star} (P^{\pi^\star})^j \right] (s_t, a_t)}{\mu_{\mathsf{b}} (s_t, a_t)} \sum_{i=1}^{n_t(s_t, a_t)} \eta_i^{n_t(s_t, a_t)} P_{s_t, a_t} \left( V^\star - V_{k_i(s_t, a_t)} \right) \right]$$

$$= \sum_{t=t_{\mathsf{mix}}(\delta)}^{T} \left[ \sum_{s \in \mathcal{S}, a \in \mathcal{A}} \widetilde{d}(s, a) \sum_{i=1}^{n_t(s,a)} \eta_i^{n_t(s,a)} P_{s,a} \left( V^\star - V_{k_i(s,a)} \right) \right.$$

$$\left. - \left( 1 + \frac{1}{\Gamma} \right) \frac{\widetilde{d}(s_t, a_t)}{\mu_{\mathsf{b}} (s_t, a_t)} \sum_{i=1}^{n_t(s_t, a_t)} \eta_i^{n_t(s_t, a_t)} P_{s_t, a_t} \left( V^\star - V_{k_i(s_t, a_t)} \right) \right].$$

Here,

$$\widetilde{d}(s, a) := \sum_{j=0}^{\infty} \gamma \left[ \gamma \left( 1 + \frac{1}{\Gamma} \right)^3 \right]^j \left[ \rho^{\pi^\star} (P^{\pi^\star})^j \right] (s, a)$$

for any $(s, a) \in \mathcal{S} \times \mathcal{A}$. Note that this equation exactly matches with Step 2.4 in (Yan et al., 2022), hence the remaining proof similarly follows, and is omitted here. Specifically, we have that

$$\psi \lesssim \frac{C^\star t_{\mathsf{mix}} \iota}{(1-\gamma)^3} \log^2 \left( \frac{T}{\delta} \right) + \frac{C^\star S t_{\mathsf{mix}}}{(1-\gamma)^2} \log \left( \frac{T}{\delta} \right).$$

**Bounding $\phi$.** Similar to (Yan et al., 2022), we can employ an analogous argument to show that $\phi$ can be bounded as

$$\phi \lesssim \frac{C^\star t_{\mathsf{mix}} \iota}{(1-\gamma)^3} \log^2 \left( \frac{T}{\delta} \right) + \frac{C^\star S t_{\mathsf{mix}}}{(1-\gamma)^2} \log \left( \frac{T}{\delta} \right).$$

Now, plugging the bounds on $\alpha$, $\theta$, $\psi$ and $\phi$ further implies that

$$\alpha_0 \leq \alpha + \xi + \theta + \psi + \phi$$

$$\lesssim \sqrt{\frac{C^\star S T \iota^2}{(1-\gamma)^5}} + \frac{C^\star S t_{\mathsf{mix}} \iota}{(1-\gamma)^2} + \frac{C^\star t_{\mathsf{mix}} \iota}{(1-\gamma)^3} \log^2 \left( \frac{T}{\delta} \right).$$

We then invoke equation 133 to conclude that

$$V^{\star}\left(\rho\right) - V^{\widehat{\pi}}\left(\rho\right) \leq \frac{\alpha_0}{T} \lesssim \sqrt{\frac{C^{\star}S\iota^2}{T\left(1-\gamma\right)^5}} + \frac{C^{\star}St_{\mathsf{mix}}\iota}{T\left(1-\gamma\right)^2} + \frac{C^{\star}t_{\mathsf{mix}}\iota^2}{T\left(1-\gamma\right)^3}.$$

This hence completes the proof. □

