# OpenReview forum: "Model-Free Offline Reinforcement Learning with Enhanced Robustness"
_ICLR.cc/2025/Conference — ICLR 2025 Poster_

### Official Review · Reviewer_sEko · 2024-11-01

**Soundness:** 3
**Presentation:** 3
**Contribution:** 3
**Rating:** 6
**Confidence:** 3

**Summary:**

The paper presents a novel approach to offline reinforcement learning (RL) by introducing a double-pessimism principle. This principle aims to enhance robustness and scalability by conservatively estimating performance, accounting for both limited data and potential model mismatches. The authors propose a model-free algorithm that is robust to environment mismatches and provide a rigorous sample complexity analysis for a specific case of mismatch modeling using the $l_\alpha$-norm. The paper also includes some experiments demonstrating the approach's effectiveness.

**Strengths:**

- The introduction of the double-pessimism principle is a significant contribution to offline RL. By addressing both data limitations and model mismatches, the approach provides a more comprehensive framework for robust policy learning.
- The paper offers a detailed sample complexity analysis for the $l_\alpha$-norm model, which theoretically demonstrates the efficiency of the proposed method. This analysis is crucial for understanding the algorithm's performance and provides a solid foundation for its claims.
- Some experiments conducted in both simulated and real environments showcase the practical applicability and robustness of the proposed algorithm. The results indicate that the double-pessimism approach outperforms existing methods in handling model uncertainty.

**Weaknesses:**

- The paper's focus on perturbations over transition probabilities is a limitation. In real-world applications, perturbations can occur in various forms, not just in transition probabilities. This narrow focus may hinder the practical applicability of the approach in more complex environments where other types of perturbations are present.
- While the sample complexity analysis is rigorous, it is limited to the $l_\alpha$-norm model. This restricts the generalizability of the theoretical findings to other types of perturbations or uncertainty models. A broader analysis covering more general cases would strengthen the paper's contributions.
- For general perturbations, the approach may require solving large optimization problems for each update. This could significantly affect learning efficiency, especially in large-scale or real-time applications. The paper would benefit from a discussion on how to mitigate these computational challenges.

**Questions:**

- Can you provide more details on the algorithm's implementation efficiency, particularly regarding the $\kappa$ calculation? What is its computational complexity?
- Does the current definition of perturbations ensure that their impacts are not cascading—meaning they don't affect future steps?

---

> ### Author Response · Authors · 2024-11-19
> **Response-part 1**
>
> We sincerely thank the reviewer for your thoughtful feedback and suggestions. We have modified our paper accordingly, with the changes highlighted in blue. Below we provide a point-by-point response to the reviewer's questions and concerns.
>
> **W1. Perturbations over transition probabilities are a limitation.**
>
> We first emphasize that uncertainties in transition probabilities are prevalent and significant in many real-world scenarios. For example, in robotic control under changing environmental conditions (e.g., friction or wind dynamics), the transition probabilities may differ between training and deployment. Similarly, in financial markets or healthcare, uncertainties in system dynamics can affect the reliability of models trained on historical data. Addressing these transition uncertainties is crucial for ensuring robust policy performance.
>
>
> On the other hand, we acknowledge that perturbations in real-world applications can manifest in various forms, not just transition probabilities. Examples include uncertainties in the reward signal [1] and observations [2], which may require distinct solutions or algorithmic approaches. However, our double-pessimism framework is versatile and can be extended to handle these additional uncertainties, which opens up opportunities for future research. For instance, in the case of reward uncertainty, a distributional uncertainty set for rewards can be constructed, introducing an additional penalty term consistent with our pessimism principle. Thus, while we concentrate on transition uncertainty, our framework is adaptable and can be extended to other types of perturbations with appropriate modifications.
>
> [1] Liang, Xinran, et al. "Reward uncertainty for exploration in preference-based reinforcement learning." ICLR (2022).
>
> [2] Pinto, Lerrel, et al. "Robust adversarial reinforcement learning." ICML (2017).
>
>
>
> **W2. Sample complexity analysis is limited to the $\ell_\alpha$-norm model.**
>
>
> While our current analysis is focused on the $\ell_\alpha$-norm model, we acknowledge that different uncertainty models possess unique characteristics, often requiring tailored analytical techniques, and it will involve a substantially broader scope, which we propose as a promising direction for future research.
>
> Nonetheless, our theoretical analysis can be extended to other uncertainty models. For example, in the Appendix, we illustrate how to design the model mismatch penalty term $\kappa$ under the $\chi^2$ divergence model. By leveraging similar principles used in our $\ell_\alpha$-norm analysis, as well as the results provided for both models in [1], we anticipate achieving comparable sample complexity bounds. This adaptability highlights the broader applicability of our framework, making it a versatile tool for addressing robustness in offline reinforcement learning under various types of model perturbations.
>
> [1] Shi, Laixi, et al. "The curious price of distributional robustness in reinforcement learning with a generative model." NeurIPS (2024).
>
>
>
> **W3 \& Q1. For general perturbations, how to mitigate the computational challenges.**
>
> We agree that our approach involves solving an optimization problem for each update. However, for the uncertainty models we considered, including the $\ell_\alpha$-norm and $\chi^2$ divergence, these optimization problems generally yield closed-form solutions, ensuring efficient computation (see the case study in Appendix C.2 of the updated paper). The computational complexity in these cases is $\mathcal{O}(S)$, resulting in a per-step complexity of at most $\mathcal{O}(SA)$ for our model-free algorithm. This is notably more efficient than model-based methods, which have a computational cost of $\mathcal{O}(S^2A)$ per step due to the need to estimate and update the entire transition model. Thus, our approach  improves  the computational efficiency. We have included a detailed discussion on computational costs in our paper.
>
> For other types of uncertainty sets, such as those based on KL divergence or Wasserstein distance, we first need to confirm and specify their closed-form solutions, which may be derived through duality techniques [1, 2]. If they do not admit a closed-form solution, numerical optimization methods would be required.
>
> [1] Panaganti, Kishan, and Dileep Kalathil. "Sample complexity of robust reinforcement learning with a generative model." ICML 2022.
>
> [2] Gao, Rui, and Anton J. Kleywegt. "Distributionally robust stochastic optimization with wasserstein distance." arXiv preprint arXiv:1604.02199.

---

> ### Author Response · Authors · 2024-11-19
> **Response-part 2**
>
> **Q2. Does the current definition of perturbations ensure that their impacts are not cascading—meaning they don't affect future steps?**
>
> The current definition of perturbations allows for cascading impacts, as these perturbations influence the underlying transition dynamics, thereby affecting subsequent state transitions. This means that a single perturbation can lead the environment into states that may have long-lasting effects on future decisions. For example, a perturbation could cause the system to transition into a low-reward state, from which it might require multiple steps to recover. Thus, the impact of perturbations is not confined to a single step but can propagate throughout the trajectory, affecting long-term performance.
>
> This cascading effect is a deliberate aspect of our design, as our framework is specifically intended to prepare the agent for robust performance against a variety of potential adverse scenarios that may persist over multiple steps. By accounting for the long-term impact of perturbations, our approach aims to enhance robustness in environments where the effects of uncertainties are not merely transient but can accumulate and compound over time.

---

> ### Author Response · Authors · 2024-11-22
>
> We sincerely thank the reviewer for taking the time and effort to evaluate our manuscript. If there are any remaining concerns, please do not hesitate to let us know as the discussion period draws to a close.

---

### Official Review · Reviewer_w3Hx · 2024-11-02

**Soundness:** 3
**Presentation:** 2
**Contribution:** 3
**Rating:** 6
**Confidence:** 3

**Summary:**

The paper presents a new approach for offline reinforcement learning that emphasizes both robustness to model mismatch and limited dataset. The paper provides theoretical results regarding robust value estimation, sub-optimality gap, and sample efficiency. The empirical results also support the superiority of the proposed double-pessimism approach, compared to the single-pessimism approach.

**Strengths:**

- The paper proposed the first offline model-free algorithm with a robustness guarantee.
- The empirical results on toy environments show the proposed double-pessimism approach outperforms the single-pessimism baseline.

**Weaknesses:**

- The proposed method has a higher sample complexity than the baseline (Shi & Chi, 2022) regarding gamma.
- Adding another pessimistic term might lead to overly conservative policies, especially when the model mismatch problem is not significant.
- The source code is not provided, making it difficult to assess the reproducibility of the results.
- It would be more practical if the authors could provide a real-world example where model mismatch is likely to occur and construct the uncertainty set based on that scenario. While the mismatch modeled by the \ell_\alpha norm is suitable for theoretical guarantees, it is unclear if real-world perturbations would follow this type.

**Questions:**

- In Figure 2, can the authors provide the standard deviations or confidence intervals for the y-axis of each method? The envelopes in Figure 1 were informative, but Figure 2 does not provide such.

---

> ### Author Response · Authors · 2024-11-19
> **Response-part 1**
>
> We sincerely thank the reviewer for your thoughtful feedback and suggestions. We have modified our paper accordingly with the changes and additions highlighted in blue. Below we provide a point-by-point response to the questions and concerns.
>
>
> **W1. Higher sample complexity than the baseline.**
>
> In the RL literature, a distinction is made between model-based and model-free methods. We would like to clarify that the method in (Shi \& Chi, 2022) is **model-based**, requiring estimation and storage of the entire transition model, which incurs a space complexity of $\mathcal{O}(S^2A)$. In contrast, our **model-free** method eliminates the need to store the full transition dynamics, significantly reducing the space complexity to $\mathcal{O}(SA)$. Moreover, the implementation complexity of our model-free algorithm is also smaller than model-based ones (See the discussion in Appendix C.2). These substantial reductions enhance the scalability of our method in large-scale environments.
>
> Regarding the dependence on $\gamma$ for sample complexity, it is important to emphasize that higher sample complexity is inevitable in model-free methods, due to the absence of explicit transition model information and the use of a bootstrapping-style updating rule. Such a claim has been evidenced by studies under many different settings, for example, non-robust offline RL ([1] v.s. [2]); Non-robust RL with generative models ([3] v.s. [4,5]). Even with advanced techniques like variance reduction, model-free approaches still incur a higher sample complexity [6]. Hence, we also expect that a higher complexity in our model-free method is fundamentally unavoidable.
>
>
> However, it is crucial to emphasize that our approach achieves the optimal sample complexity for model-free algorithms under the non-robust setting. Thus, our method achieves the best possible trade-off between sample efficiency and space complexity, and improves the scalability of offline robust RL.
>
>  [1] Li, Gen, et al. "Settling the sample complexity of model-based offline reinforcement learning." The Annals of Statistics 52.1 (2024): 233-260.
>
>  [2] Shi, Laixi, et al. "Pessimistic q-learning for offline reinforcement learning: Towards optimal sample complexity." ICML, 2022.
>
>  [3] Azar, Mohammad Gheshlaghi, Rémi Munos, and Bert Kappen. "On the sample complexity of reinforcement learning with a generative model." arXiv preprint arXiv:1206.6461 (2012).
>
>  [4] Li, Gen, et al. "Breaking the sample size barrier in model-based reinforcement learning with a generative model." NeurIPS 2022.
>
>  [5] Li, Gen, et al. "Sample complexity of asynchronous Q-learning: Sharper analysis and variance reduction." NeurIPS 2020.
>
>  [6] Li, Gen, et al. "Is Q-learning minimax optimal? a tight sample complexity analysis." Operations Research 72.1 (2024): 222-236.
>
>  **W2. Adding another pessimistic term might lead to overly conservative policies, especially when the model mismatch problem is not significant.**
>
> We agree that in some cases, model mismatch may not be significant, and we identify two such scenarios. However, we argue that the potential over-pessimism from the additional penalty term can be effectively mitigated under both cases.
>
> Firstly, when the model mismatch itself is small (i.e., the dataset closely resembles the deployment environment), a large pessimism term might result in overly conservative policies. This nevertheless can be avoided since the learner controls the penalty term via the uncertainty set radius $\Gamma$, which measures the model mismatch. If the learner knows or believes the mismatch is minimal, a smaller $\Gamma$ can be chosen to reduce the penalty and prevent unduly pessimism. And if no such prior knowledge is available, the suitable choice of $\Gamma$ can be decided through trial-and-error methods or grid search.
>
>
> The second case is that, although model mismatch is significant, the agent's performance is not sensitive to environmental perturbations, and optimal policies across different models might remain similar. In this scenario, the additional pessimism will not be overly conservative, even with a larger value of $\Gamma$. This is because the worst-case scenario will approximate the true deployment environment, since the performance is not sensitive to environmental deviations. Thus, the optimal robust policy learned through the double-pessimism principle will still align closely with the true optimal policy, naturally alleviating pessimism and preventing overly conservative behavior.

---

> ### Author Response · Authors · 2024-11-19
> **Response-part 2**
>
> **W3. The source code is not provided, making it difficult to assess the reproducibility of the results.**
> We will provide the code upon paper acceptance. Our primary focus is on developing the framework with rigorous theoretical guarantees, particularly within the tabular setting. Given the straightforward nature of our experiments, which aim to validate theoretical insights, we initially opted not to include the code. However, we are committed to sharing it to ensure the reproducibility of our results.
>
>
>
> **W4. Provide a real-world example where model mismatch is likely to occur and construct the uncertainty set based on that scenario. Will the real-world perturbations follow $l_\alpha$-norm structure?**
>
>
> Model mismatch is a common challenge in real-world applications, also known as the Sim-to-Real Gap. In some simulation-based problems like video games, the transition dynamics can be directly controlled and the uncertainty can be captured through a $l_\alpha$-norm uncertainty on the transition probabilities. In practical problems, more complicated uncertainty sets can be constructed for model mismatch. For example, in UAV control systems, transition dynamics can vary significantly due to changing weather conditions and can result in model mismatch. We can then construct an uncertainty set by leveraging trajectory prediction and domain knowledge, like fluid mechanics, to account for these variations. Similarly, in robot manipulation tasks, robots are trained in simulated environments and deployed in different real-world settings, and we can construct an uncertainty set by perturbing the training field conditions to bridge the mismatch gap between the training field and potential deployment scenarios.
>
>
> In real-world tasks, the transition dynamics resulting from environmental perturbations can be quite complicated, and may not be perfectly captured by any simple-structured uncertainty sets. Our uncertainty set is an idealistic and simplified model to constrain the amplitudes of the perturbations. However, our experiments on classic control problems and one D4RL benchmark (see Appendix B) demonstrate that implementing our algorithm with the $\ell_\alpha$-norm uncertainty set still enhances robustness, illustrating the effectiveness of our framework. This enhancement comes from our model's ability to capture a meaningful subset of possible environmental variations, allowing our pessimism principle to improve robustness even with approximate uncertainty modeling.
>
>
> **Q1.  In Figure 2, can the authors provide the standard deviations or confidence intervals for the y-axis of each method?**
>
> We have included the standard deviations for all experimental results in Figure 2. As shown, our double-pessimism algorithm consistently outperforms the baseline in terms of robustness across all tested scenarios, enjoying more stable deviations and demonstrating robustness.

---

> ### Author Response · Authors · 2024-11-22
>
> We sincerely thank the reviewer for taking the time and effort to evaluate our manuscript. If there are any remaining concerns, please do not hesitate to let us know as the discussion period draws to a close. If our responses have satisfactorily addressed your concerns, we kindly request you to consider revising your score.

---

> ### Comment · Reviewer_w3Hx · 2024-11-24
>
> Thank you for the detailed response. The responses address most of my concerns, and I have raised my score accordingly.

---

> > ### Author Response · Authors · 2024-11-24
> >
> > Thank you for your response! We are happy the revisions addressed your concerns and are open to any further discussion.

---

### Official Review · Reviewer_cMFi · 2024-11-04

**Soundness:** 4
**Presentation:** 3
**Contribution:** 4
**Rating:** 6
**Confidence:** 3

**Summary:**

This paper presents a model-free offline RL algorithm using a "double-pessimism principle" to enhance robustness against model mismatches and scalability, effectively handling uncertainties from limited data and environment shifts.

**Strengths:**

1. The paper is well-structured.
2. The double-pessimism principle effectively tackles uncertainties from both data limitations and environment mismatch, adding a valuable perspective to offline RL.
3. By avoiding transition model estimation, the algorithm shows strong scalability potential for complex environments.
4. The paper provides a detailed sample complexity analysis and shows performance improvements on benchmark tasks, supporting the method's effectiveness.

**Weaknesses:**

1. While the paper shows strong results, additional tests on more diverse, high-dimensional environments could provide a clearer picture of scalability, such as d4rl.
2. Comparing other offline RL algorithms, such as CQL, IQL, etc., will further enhance the contribution of this work.

**Questions:**

n/a

---

> ### Author Response · Authors · 2024-11-19
>
> We sincerely thank the reviewer for your thoughtful feedback and suggestions. We have modified our paper accordingly, with changes highlighted in blue.
>
> We want to highlight that the primary objective of our paper is to develop a framework for offline RL with model mismatch with a rigorous theoretical foundation, and hence we focus on the tabular case to develop precise justifications and analysis. While we initially used simple environments as proof of concept, we acknowledge the importance of broader experimental validation. Therefore, we have added comprehensive experimental results in the appendix to demonstrate our method's effectiveness across different settings.
>
> Below we provide a response to each specific comment.
>
> **Additional tests on more diverse, high-dimensional environments. Compare with other offline RL algorithms.**
>
> First, we conducted experiments in three relatively more complex tabular environments in Appendix B.1, B.3 and C.3 of the updated paper (Frozen-Lake and Taxi environments from OpenAI Gym, and the American Option problem), all of which demonstrate the effectiveness of our framework, particularly in improving robustness.
>
>
> As for the comparison with other offline RL algorithms, we first emphasize that these algorithms are typically designed for large-scale offline RL settings **without model mismatch**, and are also expected to be outperformed by ours when applied to the tabular setting with model mismatch.
>
> On the other hand, these algorithms achieve scalability by approximating the $Q$-function using neural networks, making them more efficient than directly applying our tabular algorithm, but we can also incorporate function approximation into our framework to improve scalability.  To illustrate the adaptability and effectiveness of our framework in larger-scale problems, we develop a double-pessimism CQL algorithm, as detailed in Appendix B.2. Specifically, we utilize the function approximation technique and the pessimism principle for dataset of the CQL algorithm, and incorporate an additional layer of pessimism to enhance robustness against model mismatch. Comparisons between our double-pessimism CQL and vanilla CQL under CartPole from OpenAI Gym and halfcheetah-medium-v2 environment from D4RL demonstrate significant improvements in robustness against model uncertainty, further validating the effectiveness of our framework.
>
> It is also of interest to evaluate the effectiveness of our approach on other benchmarks of D4RL and other high dimensional ones. However, this may be beyond the scope of the discussion period and this paper, which primarily focuses on establishing the theoretical foundations of our framework.

---

> ### Author Response · Authors · 2024-11-22
>
> We sincerely thank the reviewer for taking the time and effort to evaluate our manuscript. If there are any remaining concerns, please do not hesitate to let us know as the discussion period draws to a close.

---

### Official Review · Reviewer_V4q8 · 2024-11-04

**Soundness:** 4
**Presentation:** 4
**Contribution:** 3
**Rating:** 8
**Confidence:** 3

**Summary:**

In this paper, the authors proposed a novel double-pessimism principle framework for offline reinforcement learning, which conservatively estimates the performance of offline policy learning and accounts for both the pessimism caused by limited data and potential model mismatch. Compared with previous methods that usually leverage model-based approaches, the authors proposed a model free algorithm to learn the optimal policy that is robust to potential environment mismatches, which enhances robustness in a scalable manner. The authors also provide sample complexity analysis of the algorithm and extensive experiments to further demonstrate that the approach is robust and scalable.

**Strengths:**

- The authors present a scalable and robust model-free algorithm to quantify both the uncertainty of model mismatch and limited dataset; the way of constructing the model mismatch pessimistic allows the authors to derive a model-free algorithms;
- The authors present the sample complexity result for both finite horizon case and infinite discounted case.
- The authors demonstrate that with the double pessimism principle, empirically we can learn better policies compared with single pessimistic based algorithms.

**Weaknesses:**

-My central question about the paper is that in practise how we can justify the uncertainty set in real case, and how we can estimate the parameters of the uncertainty set instead of directly assigning some upper hyperparameter.

**Questions:**

- In some environments such as mujoco, how can we identify what kind of uncertainty set we should use here, and how can we estimate the uncertainty set hyperparameters to ensure the policy we learn is under the real settings?

---

> ### Author Response · Authors · 2024-11-19
>
> We sincerely thank the reviewer for your feedback and suggestions. We have modified our paper accordingly, with the changes and additions highlighted in blue. Below we provide a point-by-point response to the questions and concerns.
>
>
> **How to justify the uncertainty set in real case?  What kind of uncertainty set should we use, and how to estimate the parameters of the uncertainty set?**
>
> We introduce an uncertainty set to address potential uncertainty in the environmental dynamics. Such environmental uncertainty can be modeled through direct perturbations of the transition probabilities, which fits well in many applications.  For example, in UAV control systems, transition dynamics can be perturbed due to changing weather conditions. Similarly, in robot manipulation tasks, the dynamics of robots are also subject to environmental conditions that may not be fully captured in the training field.
>
>
> Concerning which model to use (e.g., to use $l_\alpha$-norm or KL-divergence), this may not be always easily determined in practice. In tasks like MuJoCo, the transition dynamics resulting from environmental perturbations can be quite complex, and may not be perfectly captured by any simple-structured uncertainty sets. However, our experiments on classic control problems and one D4RL benchmark (see Appendix B) demonstrate that implementing our algorithm with the $\ell_\alpha$-norm uncertainty set still enhances robustness. This enhancement comes from our model's ability to capture a meaningful subset of possible system variations, allowing our pessimism principle to improve robustness even with approximate uncertainty modeling.
>
> Regarding parameter estimation, we first note that the choice of the radius impacts the robustness of the policy: larger values increase robustness but may lead to conservative behavior, while smaller values allow more optimistic policies but with less protection against model mismatch. There are several practical methods to set the radius. Domain experts can estimate uncertainty bounds based on their knowledge of system variations and limitations, based on which the uncertainty set radius can be set. Alternatively, a data-driven approach using concentration inequalities on a small deployment dataset can statistically estimate the uncertainty radius. When these approaches are not feasible, empirical tuning through grid search or trial-and-error can optimize robustness in deployment scenarios.

---

> ### Author Response · Authors · 2024-11-22
>
> We sincerely thank the reviewer for taking the time and effort to evaluate our manuscript. We also appreciate your support of our paper. If there are any remaining concerns, please do not hesitate to let us know.

---

### Official Review · Reviewer_AZ4j · 2024-11-04

**Soundness:** 2
**Presentation:** 2
**Contribution:** 2
**Rating:** 6
**Confidence:** 2

**Summary:**

This work enhances offline RL by developing a scalable, robust model-free algorithm that optimizes worst-case performance within an uncertainty set. Using a double-pessimism principle to handle data uncertainty and model mismatch, the method adapts across settings without model estimation. Through theoretical analysis the author demonstrates the algorithm’s near-optimal data efficiency, advancing robustness and scalability in offline RL.

**Strengths:**

1. The paper introduces a "double-pessimism principle" that uniquely addresses both model mismatch and limited dataset coverage within a model-free framework
2. The authors provide a rigorous theoretical foundation, including sample complexity analysis and convergence guarantees, which enhances the paper's quality

**Weaknesses:**

1. The paper lacks sufficient experiments to demonstrate the effectiveness of the proposed method. For instance, there is no comparison with model-based baselines, and the evaluation environment is relatively simple.

2. There are no ablation experiments analyzing the functionality of the proposed "double-pessimism principle."

**Questions:**

1. As mentioned above, have the authors evaluated how the proposed appraoch performs in any of the D4RL environments, which are widely used in offline RL research?
2. Are there any strategies for fine-tuning parameters for specific tasks?

---

> ### Author Response · Authors · 2024-11-19
>
> We sincerely thank the reviewer for your thoughtful feedback and suggestions. We have modified our paper accordingly, with changes highlighted in blue.
>
> We want to highlight that the primary objective of our paper is to develop a framework for offline RL with model mismatch with a rigorous theoretical foundation, and hence we focus on the tabular case to develop precise justifications and analysis. While we initially used simple environments as a proof of concept, we acknowledge the importance of broader experimental validation. Therefore, we have added comprehensive experimental results in the appendix to demonstrate our method's effectiveness across different settings.
>
> Below we provide a response to each specific comment.
>
> **W1 \& Q1. The paper lacks sufficient experiments to demonstrate the effectiveness of the proposed method.**
>
> While our paper initially focused on tabular cases to establish theoretical foundations, we have expanded our experimental validation. First, we have added comprehensive experiments in more complex tabular settings (Appendix B.1 and C.3) in the Frozen-Lake and Taxi environments from OpenAI Gym, and the American Option problem, which consistently demonstrate our framework's effectiveness across different scenarios and uncertainty levels.
>
> On the other hand,  our **algorithms** are designed for the tabular setting, and it can be challenging to directly apply them to larger-scale benchmarks. However, we emphasize that our **double-pessimism framework** is not inherently limited to the tabular setting. In fact, it can be adapted for larger problems through the use of function approximation. To demonstrate the scalability of our framework, we have integrated it with CQL, and a detailed discussion of this adaptation is provided in Appendix B.2 of the updated paper, along with additional experimental results under CartPole from OpenAI Gym and halfcheetah-medium-v2 environment from D4RL. Our results show that by incorporating our model-mismatch pessimism principle with CQL's existing dataset uncertainty mechanism, we achieve notably improved robustness against environment mismatch. These results further demonstrate the effectiveness of our framework, which can effectively enhance robustness with function approximation techniques. We will continue evaluating our framework under other high-dimensional benchmarks in D4RL, and will include results in the final version.
>
>  **W2. There are no ablation experiments analyzing the functionality of the proposed "double-pessimism principle."**
>
> Our double-pessimism principle addresses two distinct sources of uncertainty:
> 1. Limited dataset coverage (common in offline RL);
> 2. Model mismatch between training and deployment environments (our key contribution).
>
> In Appendix B.3 of the updated paper, we present comprehensive ablation experiments comparing four variants: Vanilla Q-learning (no pessimism),
> Robust Q-learning (model-mismatch pessimism only),
> Offline non-robust Q-learning (dataset pessimism only),
> Our offline robust Q-learning (double pessimism).
> The results demonstrate that our double-pessimism approach consistently outperforms all single-pessimism and no-pessimism baselines, validating the necessity of addressing both uncertainty sources. Also, comparisons with vanilla Q-learning showcases the effectiveness of both pessimism principles.
>
>  **Q2. Are there any strategies for fine-tuning parameters for specific tasks?**
>
> Our algorithm has two key parameter types:
>
> (a). Uncertainty Set Radius ($\Gamma$). The choice of $\Gamma$ directly impacts policy robustness: larger values increase robustness but may lead to conservative behavior, while smaller values allow more optimistic policies but with less protection against model mismatch. It can be chosen through:
> (a.1) Can be set by domain experts based on expected model discrepancies between training and deployment environments;
> (a.2) Can be empirically estimated using a small deployment environment dataset, as demonstrated in [1,2], where the radius is calibrated based on observed state-transition differences.
>
>
> (b). Standard RL Parameters (learning rate, etc.):
> (b.1) Can be tuned via traditional methods like grid search and trial-and-error, with cross-validation on a held-out portion of the offline dataset;
> (b.2) Can leverage automated hyperparameter optimization techniques such as Bayesian optimization for more efficient parameter search;
> (b.3) Can incorporate domain-specific prior knowledge from similar tasks or environments to narrow down the search space.
>
> [1] Wang, Yue, Jinjun Xiong, and Shaofeng Zou. "Achieving the Asymptotically Optimal Sample Complexity of Offline Reinforcement Learning: A DRO-Based Approach." TMLR 2024.
> [2] Panaganti, Kishan, et al. "Bridging distributionally robust learning and offline rl: An approach to mitigate distribution shift and partial data coverage." arXiv preprint arXiv:2310.18434 (2023).

---

> ### Author Response · Authors · 2024-11-22
>
> We sincerely thank the reviewer for taking the time and effort to evaluate our manuscript. If there are any remaining concerns, please do not hesitate to let us know as the discussion period draws to a close. If our responses have satisfactorily addressed your concerns, we kindly request you to consider revising your score.

---

> > ### Comment · Reviewer_AZ4j · 2024-11-24
> >
> > Thanks for the reply. Most of my concerns are addressed, and I'd like to increase the score accordingly.

---

> > > ### Author Response · Authors · 2024-11-24
> > >
> > > Thank you for your response! We are happy the revisions addressed your concerns and are open to any further discussion.

---

### Author Response · Authors · 2024-11-21
**General response**

Dear Reviewers and AC,

 We would like to thank the reviewers for your constructive and insightful comments which greatly improved the quality and presentation of this paper. In this revised paper, we have carefully addressed all the comments raised and have updated the paper in accordance with them, highlighted in blue.

The primary objective of our paper is to establish a rigorous theoretical framework for offline RL with model mismatch, focusing on the tabular setting to ensure precise justifications and analysis. Our complexity results represent the first ones for model-free offline robust RL, and matches the optimal complexity for model-free non-robust offline RL. We appreciate the reviewers' recognition of the strength of our theoretical analysis.

On the other hand, as some reviewers suggested, we develop more comprehensive experiments to verify the effectiveness and scalability of our framework. Below is a summary of the revisions made in response to the common concerns:

(1). As our algorithm is designed for tabular setting, we developed more  experimental results under more complex tabular settings (Frozen-Lake, Taxi and American Option problems) to demonstrate the effectiveness of the proposed method across various scenarios. We also include ablation experiment to verify the effectiveness.  (Appendix B.1, B.3)

(2).  Although our model-free framework is scalable under tabular settings, it can be directly extended to even larger problems by employing function approximation techniques and more efficient algorithm design. To verify this,  we utilize neural network approximation and fitted Q-learning to develop a double-pessimism CQL algorithm. Experiments under OpenAI Gym and MuJoco environments show the improved robustness. (Appendix B.2)

(3). We also discussed how to determine uncertainty sets and parameters in real-world applications.

  We greatly value the reviewers' insights and comments, and look forward to discussing with you for further valuable suggestions.

---

### Meta-Review · Area_Chair_R6Vb · 2024-12-29

**Metareview:**

This paper proposes a model-free algorithm to implement "double-pessimism" principle for offline RL, which conservatively take both limited data and potential model mismatches into account. The authors provides sample complexity analysis and simple experiments on synthetic environments.

The major issue raised by reviewers lie in the limited practical and theoretical implication: the proposed algorithm and theoretical analysis only works for tabular setting with l_\infty norm perturbation, which is difficult to be generalized for practical applications.

Moreover, as a theoretical paper, the discussion on \gamma dependence is incomplete.

**Additional Comments On Reviewer Discussion:**

In the rebuttal period, the authors provided more discussion about the aforementioned drawbacks. However, due to the essential difficulty, the concern raised by most of the reviewers about the extension of the algorithm beyond l_\infty norm is still unaddressed.

---

### Decision · Program_Chairs · 2025-01-22

Accept (Poster)